# Probing three-dimensional mesoscopic interfacial structures in a single view using multibeam X-ray coherent surface scattering and holography imaging

Miaoqi Chu [1] ✉, Zhang Jiang[1], Michael Wojcik[1], Tao Sun [1,3], Michael Sprung [2] & Jin Wang [1] ✉

Visualizing surface-supported and buried planar mesoscale structures, such as nanoelectronics, ultrathin-film quantum dots, photovoltaics, and heterogeneous catalysts, often requires high-resolution X-ray imaging and scattering. Here, we discovered that multibeam scattering in grazing-incident reflection geometry is sensitive to three-dimensional (3D) structures in a single view, which is difficult in conventional scattering or imaging approaches. We developed a 3D finite-element-based multibeam-scattering analysis to decode the heterogeneous electric-field distribution and to faithfully reproduce the complex scattering and surface features. This approach further leads to the demonstration of hard-X-ray Lloyd's mirror interference of scattering waves, resembling dark-field, high-contrast surface holography under the grazing-angle scattering conditions. A first-principles calculation of the single-view holographic images resolves the surface patterns' 3D morphology with nanometer resolutions, which is critical for ultrafine nanocircuit metrology. The holographic method and simulations pave the way for single-shot structural characterization for visualizing irreversible and morphology-transforming physical and chemical processes in situ or *operando*.

Surface/interface phenomena associated with mesoscale and low-dimensional materials are of great interest to scientists and engineers for establishing structure–function relationships. These systems include but are not limited to, planar nanoelectronic circuits[1], hierarchical mesoscale surface structures[2], thin-film-based quantum dots[3] and photovoltaic[4], heterogeneous catalysts[5], biological membranes, and supramolecules[6–8], where observation in situ, *operando*, and in real-time is important. In addition, scientific and technological applications of surface/interface fabrication and manipulation require precise determination of mesoscale 3D structures from tens micrometers down to sub-nanometers (sub-nm) in heterogeneous

and non-periodical systems that control the functionality of the devices.

Probing structures and dynamics with nanometer (nm) spatial resolution usually requires X-ray scattering and imaging techniques, thanks to the short wavelength. The Bragg diffraction from crystals[9] and small-angle scattering[10] from non-periodical materials are usually analyzed by assuming kinematic scattering using Born approximation (BA) under weak perturbations. This approach has been proven powerful in non-destructive structural characterization for vast scientific and technological applications[11,12]. With the advent of coherent X-ray sources provided by modern storage-ring-based synchrotron[13] and

[1]X-ray Science Division, Argonne National Laboratory, Lemont, IL 60439, USA. [2]Deutsches Elektronen-Synchrotron (DESY), Notkestr. 85, 22607 Hamburg, Germany. [3]Present address: Department of Materials Science and Engineering, University of Virginia, Charlottesville, VA 22904, USA. ✉e-mail: mqichu@anl.gov; wangj@anl.gov

linear-accelerator-based X-ray free-electron laser facilities[14], lensless X-ray coherent diffraction imaging (CDI) has become a popular microscopic method in materials and biological sciences[11,15]. For isolated objects such as single nanocrystalline particles, the Bragg diffraction-based CDI (BCDI)[16,17] can probe 3D strain distribution and dynamics during structural transformation[18–20]. Ptychographic X-ray computed tomography is effective in reconstructing extended samples such as integrated circuits[1,21–23] and neuronal reconstruction[24]. Thanks to the weak interaction between hard X-rays and most of the materials on nano, micro, and even macro scales, conventional transmission and Bragg CDI reconstruction methods can take full advantage of kinematic analysis (or BA), as the 3D structure can be resolved but requires scanning numerous projection angles. The reconstruction algorithm uses iterative Fourier transforms (FT) and inverse FTs, operated over the form factors of the subject in real and reciprocal spaces[11]. As a result, the X-ray coherent imaging methods, including theory, instrumentation, and applications, have been growing rapidly[25–28].

For probing surface/interface structures, the transmission scattering geometry will not be applicable. For decades, surface X-ray scattering techniques have been developed to interrogate substrate-supported structures, owing to their sensitivity to buried interfaces and the easiness of adapting in situ sample environments[29,30]. For example, X-ray reflectivity can reveal the electron density profile in the direction perpendicular to a thin film with a sub-nm resolution[31,32]. And grazing-incident small- and wide-angle X-ray scattering (GISAXS/WAXS) provides thin films' in-plane and off-the-plane structural information in either crystalline or non-crystalline forms[33,34]. Taking advantage of high-coherence X-ray sources, the coherent surface scattering imaging (CSSI) technique[35] provides ideal tools to reconstruct surface/interface structures in 3D using scattering data in the kinematic regime in grazing-incidence geometry.

In CDI and ptychography, specimens are either free of or isolated from substrates. To accommodate transmission measurement, substrate-supported samples, such as integrated circuits, must be prepared with intrusive micromachining processes. However, the supporting substrates of mesoscale planar structures are often an integral part of the sample for enabling functionalities, which can also be used to enhance the measurements, as demonstrated in this work. The grazing-incidence X-ray surface scattering offers distinct advantages by allowing more convenient and non-invasive sample preparation than transmission electron and X-ray microscopy[36]. Unlike in transmission geometry, at surfaces and interfaces, the interference between the incident, scattered, and reflected X-rays can generate multiple scattering scenarios, such as standing waves[37] and guided surface waves[38–41], which can be harnessed to provide the depth sensitivity for revealing buried structures with nm or even sub-nm spatial resolution. However, such strong surface-induced multiple scattering, for example, total external reflection (TER), analogous to total internal reflection in optical fibers, can make the surface scattering complex and render the conventional kinematic approximation ineffective. Fundamentally, quantitative analysis of X-ray (or radiations of other wavelengths) scattering at surfaces and interfaces for determining 3D mesostructures requires solving Maxwell's equations with complex boundary conditions. To understand surface scattering in the conventional X-ray scattering regime, Sinha et al. developed distorted-wave Born approximation (DWBA) to solve surface roughness effects statistically, as reported in their 1988 seminal work, which has made an enormous impact over more than 30 years[42]. Jiang et al. extended the DWBA method to situations where the surface and interface structures significantly alter the distribution of electric fields at surfaces[43], in a model-dependent fashion for resolving the statistical structure information in the sample's lateral directions. In the case of CSSI, the scattering and reflection waves near the TER in either or both incident and exit directions generate dominating multibeam scattering.

Specifically, speckles and their intensity distributions are significantly distorted near the critical angle of the TER, which is, in turn, sensitive to the depth-dependent structural characteristics in 3D. With surface patterns minimally perturbing the electric field intensity profile in the in-plane direction, Yang and Sinha developed a rigorous algorithm to reconstruct the 3D sample structures using the DWBA approach[44], where an in-plane homogeneous electric field is assumed. On the other hand, in the CSSI measurements, the size of the coherent X-ray probe is comparable with the sample dimension. The samples consisting of high-density materials can significantly alter the electric field distribution in the off-the-plane direction and in the plane locally. As a result, the average lateral electric field assumption in the conventional DWBA becomes insufficient, even after considering the partial coherence effects[45]. Thus, the analysis assuming an in-plane homogeneous electric-field profile cannot treat the general coherent scattering cases.

On the other hand, Lloyd's-mirror interferences were observed in radio waves, lasers, and soft X-rays, with applications such as plasma density measurement and surface metrology[46–49]. Thus far, Lloyd's mirror-based holography involves the inference with the incident (or primary) waves (beams) in the bright field. Generally, conventional holography uses a plane wave or a known wavefront as the reference beam to generate the hologram and reconstruct the real-space images[50]. Due to the high coherence, sensitivities to chemical and magnetic properties, and easiness of wavefront manipulation, most of the holographic methods have been developed in soft X-ray and EUV regimes, and they rely on computational methods to retrieve the phase information to reconstruct 2D or 3D real-space images[51–57]. With hard X-rays, it is more challenging to generate small and high-flux (and coherent) plane waves as the reference for studying micro- or nano-size samples. Thus far, most hard X-ray holography has used the interference of X-ray-induced fluorescence to study single-crystal structures[58–61]. One of the recent examples is by Jiang et al. extending the fluorescence holography from a simple X-ray mirror case[62] to a one-dimensional X-ray waveguide, which resulted in characterizing nanoparticle diffusion with sub-nm resolution in real-time[63].

In this work, to overcome these difficulties of gaining true 3D morphological information from coherent surface scattering, we develop a finite-element DWBA (FE-DWBA) approach using a 3D grid system to compute the scattering from highly heterogenous surface structures in a self-consistent and accurate fashion. The FE-DWBA approach is validated by experimental data from 3D patterns in both simple and complex forms—revealing a quantitative agreement between simulation and experiment results. The FE-DWBA simulations allow us to discover the holographic nature of surface scattering in grazing-incidence geometry, arising from interference between the two most prominent surface scattering paths. This resembles Lloyd's mirror interference concept but differs distinctively from previously proposed experimental themes. The holographic imaging demonstrated here is from the interference between the scattering wave from the 3D planar pattern at the surface and its reflection from the substrate. The interfering waves, traveling along the sample/substrate horizon and significantly away from the primary and reflected X-ray beam directions, form dark-field, high-contrast, and low-noise holographic patterns that contain all 3D information of the sample morphology in a single-view image. With the surface holography concept, we demonstrate that we can unlock the 3D information accurately with a first-principles calculation using the interference of the two scattering paths with a modified free-propagation approach. The analysis is tested with a simple but true 3D sample by varying thicknesses across the surface. The holographic calculation captures all the tell-tale features of the 3D micro-patterns with unprecedented nm or sub-nm spatial resolution. Reproducing these features by a finite-element-based numerical simulation may be difficult, if not impossible, due to the computational cost and simulation limitations.

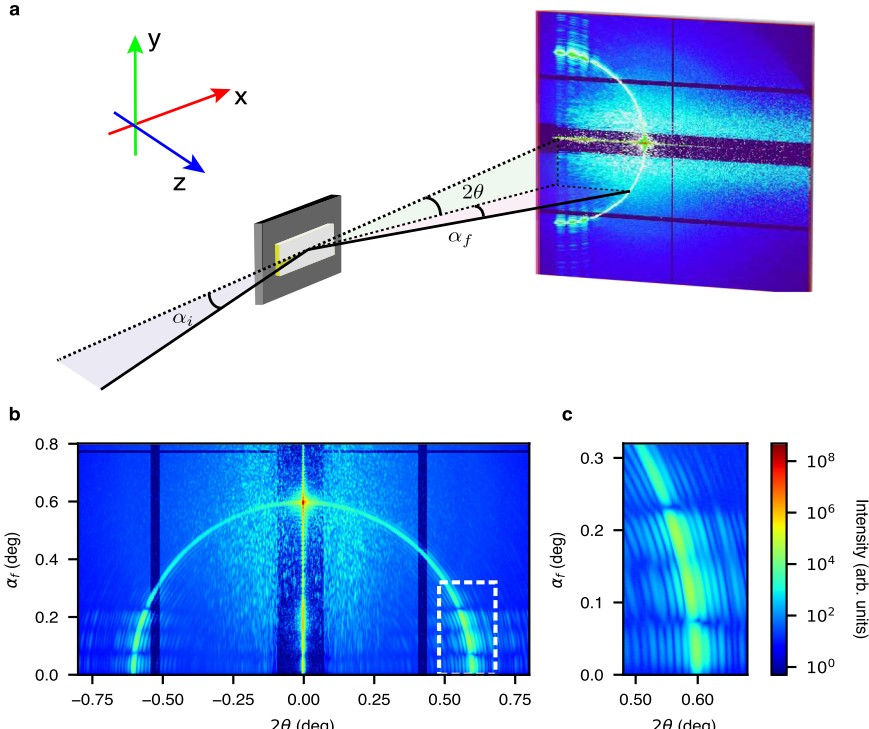

**Fig. 1 | X-ray coherent surface scattering imaging (CSSI) of an ultrathin gold single micro-bar pattern supported by a flat silicon substrate. a** Reflective scattering geometry (drawn not to scale) showing the grazing-incident angle ($\alpha_i$) of the coherent X-ray beam impinging onto the surface pattern, the reflected and scattered X-rays at an exit angle ($\alpha_f$), and azimuthal scattering angle ($2\theta$). The cartesian coordinate systems are fixed at the sample surface: x along the X-ray beam direction, y parallel to the surface and perpendicular to the beam, and z perpendicular to the substrate surface (out-of-plane). **b** CSSI scattering pattern recorded by an EIGER X4M detector placed 5 m away from the sample. The pixels are converted linearly to the scattering angles ($\alpha_f$ and $2\theta$), which are the parameters or coordinators used in most of the simulations in this work. **c** Close-up of the scattering pattern in a low-$\alpha_f$ and high-$2\theta$ region, where the intensity is dominated by multibeam scattering interference in the dashed-line box in panel **b**.

## Results

### Experiment setup

To illustrate and use the dynamical scattering effect from the sample and substrate near the critical angle, we first designed a simple sample so that all sample features and form factors are identifiable in a single scattering image at any grazing-incidence angle. The 3D surface pattern consists of an electron-beam-lithography patterned gold (Au) bar with dimensions of 4.0-μm width, 70-μm length, and 55-nm thickness, deposited onto polished silicon (Si) substrate with the 5-nm titanium (Ti) adhesive layer. The preparation of the samples is described in more detail in Methods. Coherent surface scattering imaging of this sample was performed at P10, a dedicated coherent scattering beamline at the PETRA III synchrotron facility, using an X-ray photon energy, $E$, of 8.0 keV (wavelength $\lambda = 0.155$ nm). At this wavelength, the increments of the refractive index of Si and Au are $7.67 \times 10^{-6}$ and $4.77 \times 10^{-5}$, resulting in critical angles of 0.22° and 0.56°, respectively. The monochromatic and coherent X-ray beam was focused to $1.7 \times 3.5$ (V × H, full-width-at-half-maximum, FWHM) μm² using a set of compound refractive lenses. The detector is an EIGER X 4M with $75 \times 75$ μm² pixels placed 5.0 m downstream from the sample. As shown in Fig. 1a, X-rays impinge on the sample at an incident angle of $\alpha_i = 0.6°$, above both critical angles. The coordinate system is described in more detail in Methods and Supplementary Information (SI) #1. Figure 1b shows the measured scattered intensity distribution in logarithmic scales. Due to the high coherent flux and coherence-preserving optics at the beamline, coherently scattered speckles are strong and visible on the entire detector, even with an exposure time of 1 s per image. In the higher exit angle ($\alpha_f > 0.4$) region, the scattering is mostly kinematic, as discussed in SI #1, where an overall interpretation of the coherent image is given. Briefly, the arch-shaped scattering pattern is due to how

the detector intercepts the Ewald sphere in the grazing-incidence reflection geometry[35]. The scattering pattern from the simple bar sample can be related to the sample form factors more easily when remapped from the ($\alpha_f$, $2\theta$) coordinates into the conventional reciprocal space or the $q_x - q_y$ plane using Eqs. S1–S3. The coherent scattering image near the reflection spot ($q_x = 0$ and $q_y = 0$) resembles the kinematic scattering from a rectangular pattern in transmission geometry. As shown in Fig. S1c, d, the oscillatory intensity along $q_x$ and $q_y$ directions is the signature of the pattern length (70 μm) along and the width (4 μm) perpendicular to the X-ray beam, respectively. Although the single incident angle speckle pattern contains a limited $q_z$ contribution in the kinematic regime, the oversampling from data is insufficient to reconstruct the sample structure in the z-direction. The kinematic scattering patterns taken at a series of incident angles are the base for reconstructing the surface structure in 3D, using the FT methods as demonstrated in ref. 35, which is not the subject of this work. Rather, the focus of this study is the large area in Fig. 1b (in $\alpha_f - 2\theta$ space) where $\alpha_f < 0.4°$, where the momentum transfer in the z-direction cannot be defined by Eqs. S1–S3 and a simple $q_z$ due to multibeam scattering and interference, or dynamical scattering effects. Note the intense scattering speckles in the region just above the horizon ($\alpha_f$ slightly larger than 0) well below the Si and Au critical angles. This low exit-angle region below the critical angles, from 0 to 0.3° as highlighted in the box and magnified in Fig. 1c, should not have contained intense scattering because of the optical reciprocity. Instead, the strong scattering near the sample/substrate horizon is linked to the surface waveguide phenomena[41,64] due to the substrate-induced multibeam interference, analogous to standing waves generated above an X-ray mirror below or near the critical angle, the focus of this work. Furthermore, high-intensity, high-contrast speckle fringes

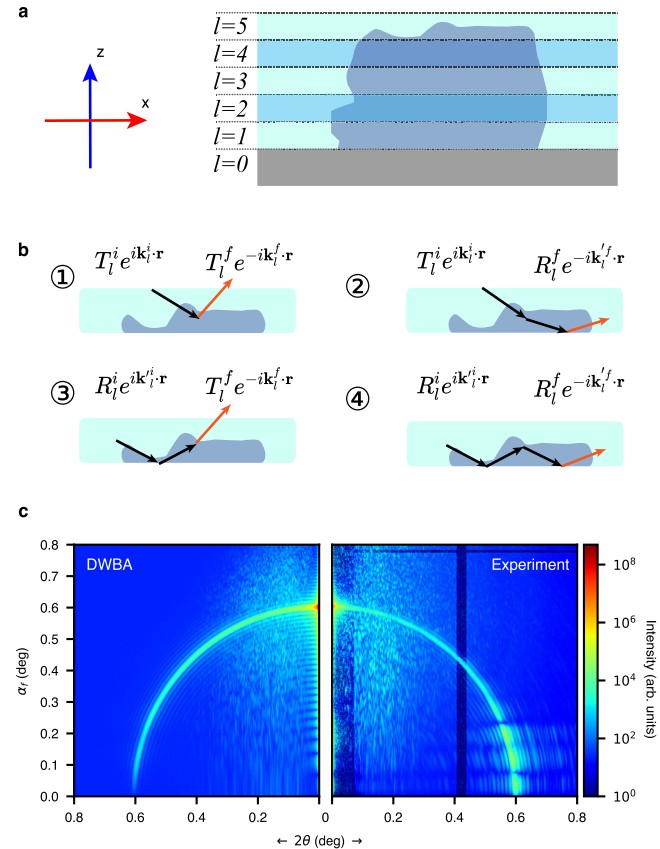

**Fig. 2 | Scattering intensity distribution simulated based on conventional distorted-wave Born approximation (DWBA). a** Schematic of conventional DWBA scattering wave vector transfers in the reflection geometry (not to scale) with a layered approach developed for evaluating grazing-incidence small-angle scattering (GISAXS) using incoherent X-rays. In the DWBA scheme, the electron density in a layer is averaged in the x-y plane over the X-ray footprint, as highlighted in the shaded areas of the schematic. **b** Schematic of the four momentum-transfer terms in the DWBA theory, which constitute the components described in Eqs. (S14–S17) in an arbitrary single layer, e.g., the top layer in the schematic. The shading colors represent the lateral electron density profile in the layer. The red arrows highlight the exit wave vector in each scattering path. **c** Surface scattering intensity, when all four terms are included in the simulation surface scattering intensity, fails to reproduce the experimental data when $\alpha_f$ is between 0° and 0.3°.

are observed along and around the arc in Fig. 1c. The periodic scattering fringes along both $\alpha_f$ and $2\theta$ angles indicate that the 3D form factors of the surface pattern can be retrieved, as shown in the following subsections.

First, we compute the expected scattering pattern using the kinematic or BA approach (as described in SI #2), assuming the electrons in the sample scatter the unperturbed incident plane wave. Thus, the scattering intensity is given by the squared modulus of the FT of the sample's electron density profile. As expected, the BA fails to predict the scattering patterns satisfactorily (Fig. S2b). Starkly different from the experimental data, the simulated scattering pattern lacks all features in the low exit angle region, where dynamical scattering effects dominate.

The DWBA is another powerful approach for computing surface scattering from flat and rough surfaces[42]. In the DWBA, thin nanostructures at a substrate can be treated as a strong perturbation to the sample's electron density and the electric field in the out-of-plane direction. The sample electron density profile in the z-direction is averaged over the in-plane directions to calculate perturbed incident and exit wavefronts. The scattering of a surface structure supported by a substrate can be summarized in Fig. 2a, the layered approach in the

out-of-plane direction. Figure 2b shows the most critical aspect of DWBA, the multibeam scattering paths, which account for four scattering components in layer $l$: (1) scattering of the incident beam by the layer, the same as BA (kinematic theory), (2) scattering from the sample then reflected at the layer bottom interface, (3) scattering from the reflected beam, and (4) reflected scattering from the reflected beam from the bottom interface. The notation describing the four scattering paths in the illustration are $T_l^{i,f}$ and $R_l^{i,f}$, the incident and exit (superscripts $i$ and $f$, respectively) complex transmissive ($T$) and reflective ($R$) coefficients, $\mathbf{k}_l^{i,f}$, incident and exit wave vectors in layer $l$, expressed in the coordinate system, $\mathbf{r}(x, y, z)$, fixed on the sample where z is normal to the layers. This method and an algorithm can be found in refs. 43,63, where GISAXS scattering patterns can be simulated perfectly for the dynamical scattering in the low-exit angle regime.

In this experiment, the Au micropattern creates a strong in-plane inhomogeneity when its lateral dimension is comparable to the coherent X-ray beam size. Also, with high-Z material Au, the averaged electron density in the pattern layer can be higher than the substrate (silicon in this case). This renders the conventional DWBA approaches ineffective, shown in Fig. 2c, demonstrating that the dynamical scattering features in the experimental data cannot be reproduced by this layered DWBA approach. It fails to produce the intense interference fringes and the complex dynamical scattering patterns where $2\theta$ is between 0 and 0.3° on the detector. Instead, the simulated pattern below 0.3° is relatively weak, implying that X-rays cannot penetrate or exit from the sample at the low angles because they are close to and below the critical angles of the substrate 0.22° and the sample layer (0.56°). Since the simulation does not yield the multibeam interference features in the low exit angle region ($\alpha_f < 0.3°$), a new method to implement the dynamical theory is required for rigorously modeling the coherent scattering from substrate-supported surface patterns quantitatively.

### Finite-element DWBA

To tackle this problem, instead of just treating the sample as a layered structure, we further divide the sample and its surroundings into a 3D grid, not only in the out-of-plane direction as in conventional DWBA but also in the in-plane directions. The top and side views of the 3D grid are shown in Fig. 3a. In other words, our approach takes into account the sample's 3D inhomogeneity inside the coherently illuminated volume. We first estimate the region of the sample/substrate illuminated by X-rays using the measured FWHM of the X-ray probe and the incident angle. The areas outside the X-ray footprint do not contribute to the scattering process and are not included in the grid to accelerate the computation. The dimensions of a grid cell are determined by the resolution of the measurement as defined by the maximal momentum transfer in each direction on the detector of more than 4 million pixels. In practice, we chose to use smaller 3D cells size of $\delta_{x,y,z} = (175 \times 5 \times 0.5)$ nm³ to avoid numerical artifacts. With this resolution-limited cell size, the in-plane electron density variation is not detectable inside each cell. Then, the electron density profile in each vertical stack of cells, a column along the sample normal direction, in the out-of-plane direction is sliced into layers, as illustrated in Fig. 3b, where the scattering terms or paths at the interface between the 3D adjacent elements are also schematically shown. The neighboring cells are labeled as subscripts $a$ and $b$. It should be noted that the electron density of the surface sample pattern is no longer averaged over each horizontal layer for computing the electric field and coefficients. Rather, they are computed locally in each grid element. Next, the transmission and reflection in each layer are calculated by using Parratt's recursive formalism to satisfy the boundary conditions for the electromagnetic field in the out-of-plane direction. The eigenstates for the incident wave ($\psi_p^i$) and the exit wave ($\psi_p^f$) of the $p^{th}$ cell can be computed as shown in SI #3.

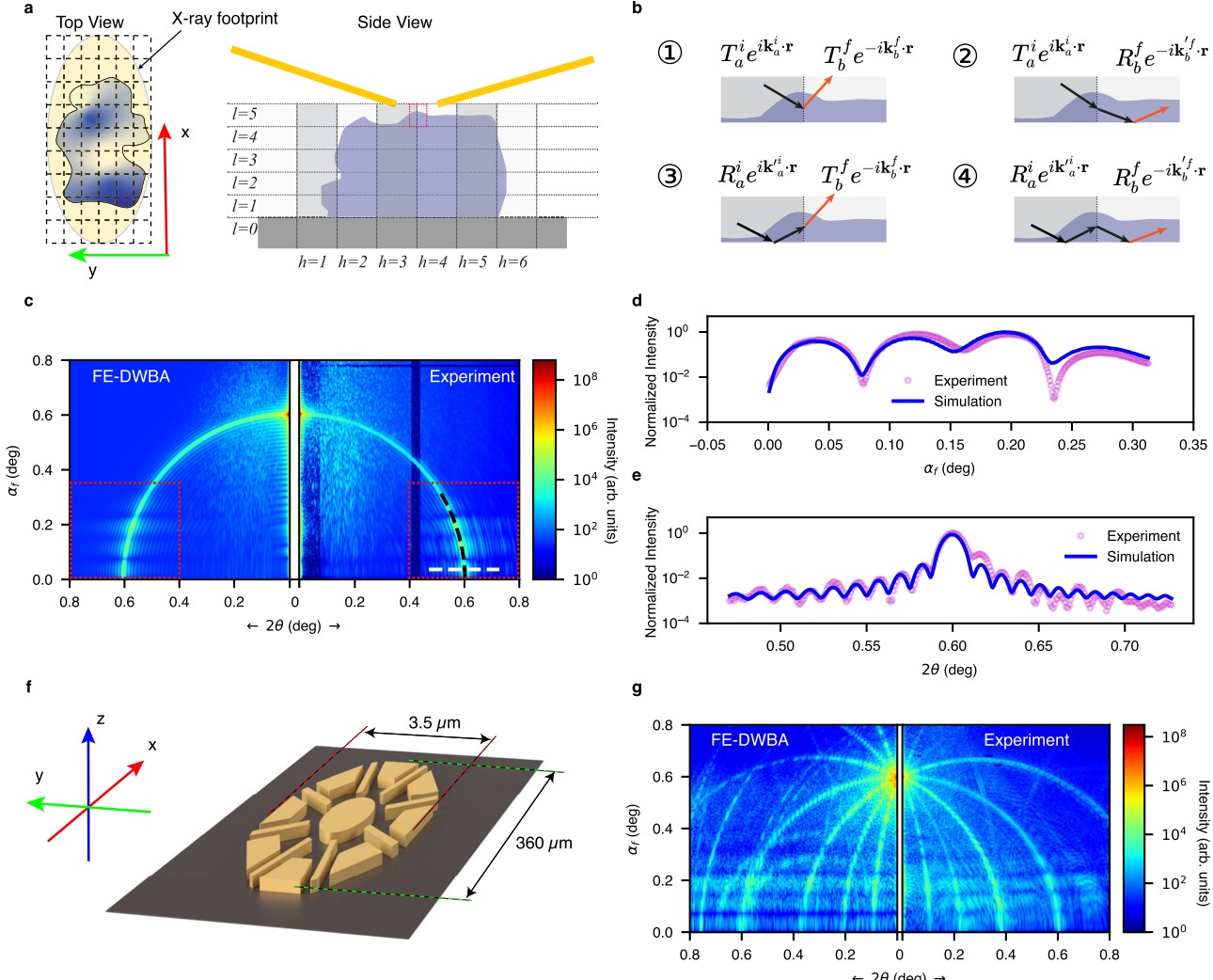

**Fig. 3 | Finite-element distorted-wave Born approximation (FE-DWBA) approach for coherent surface scattering. a** The sketch shows the top and side views of a highly heterogeneous sample and the corresponding FEA grid pattern of the substrate-supported sample (not to scale). In the top view, the shading in yellow indicates the X-ray footprint, while the gray-scale shading shows the lateral electron density variation in the surface pattern. In the side view, the shadings highlight the lateral cells in the y-direction. **b** The four scattering contributions describe the full DWBA terms in two adjacent lateral cells. The transmission and reflection coefficients have also been indexed at each lateral interface. **c** CSSI pattern calculated with the FE-DWBA algorithm. The simulated scattering pattern reproduces the low exit angle speckles and the intense interference fringes in the dashed boxes, which reflect the nature of the multibeam or dynamical scattering mechanism, revealing the pattern thickness. **d** Quantitative comparison between the FE-DWBA simulation −the line scan at the center of the arc (equivalent to $q_x = 0$) along the dashed black line from $\alpha_f = 0$ (horizon) to 0.3°, slightly above the silicon substrate critical angle (0.22°). The intensity oscillation, the signature of the dynamical scattering, below and near the critical angles of the substrate and the sample, matches well with the experimental data. **e** Scattering intensity line scan along the dashed white line containing the information on the form factor of the thin-film surface pattern along the length of the bar. The oscillation period corresponds to the bar length of 70.0 μm precisely. Therefore, this highlighted region in the single-view scattering image taken at an incident angle of 0.6° captures the 3D structural information of the sample. **f** 3D schematic of a more complex surface sample (shown not to scale), a petal-like pattern with a thickness of 55 nm. **g** Experimental data collected at 0.6° incident angle (right panel) and simulation by using the FE-DWBA method (left panel). The signature of the pattern thickness−the enhanced intensity modulations at the low exit angle is evident.

The total scattering from the grid is then the modulus square of the scattering wave within a domain from the boundaries:

$$I \propto \left| \sum_{x_p, y_p} \sum_{z_p} \left\langle \psi_p^f | \rho_p | \psi_p^i \right\rangle + \sum_{x_e, y_e \in boundaries} \sum_{z_e} \left\langle \psi_{e_1}^f | \rho_e | \psi_{e_2}^i \right\rangle \right|^2, \quad (1)$$

where the first term is the sum of all cells, which describes the dynamical scattering due to the wave distortion introduced by the layered surface structure, just like the conventional DWBA approach. The second term is the sum of cells on boundaries. $e_1$ and $e_2$ are cells downstream and upstream of cell $e$, which can vary in vertical electron density. These terms take care of the

dynamical scattering effect due to the in-plane inhomogeneity in the pattern (more in SI #3).

In terms of the FE-DWBA computational aspect, the exit eigenstates, corresponding to each pixel on the detector, need to be computed for each of the more than 5 million cells, which can make it computationally intensive. To accelerate the algorithm, the vertical stacks that share the same electron density profile are grouped. The eigenstates in each group only need to be computed once. In addition, we parallelize the computation of the scattering intensity for each exit state (Eq. 1) by using graphics processing units (GPUs). Without optimization, the total computation takes about 1 min on a single consumer-grade GPU of about 4352 cores (RTX 2080 Ti). To enable iterative reconstruction with FE-DWBA, the process can be accelerated

significantly by applying non-uniform FT to the computation of scattering (Eq. S22) and using multiple GPUs.

With the in-plane inhomogeneities incorporated in the FE-DWBA simulation, this approach generates a high-fidelity simulated scattering pattern. This is validated by the good quantitative agreement between simulation and experimental results, as shown in Fig. 3c (in the dashed-line boxes). The simulation not only resembles the overall scattering features but also produces matching positions and periods of fringes near the horizon. In addition, the simulation shows that the near-specular pattern is indeed dominated by kinematic scattering, revealing the 2D structure in the x-y plane. Figure 3d, e show scans along the black and white dashed lines in Fig. 3c, respectively. In the simulated pattern's low exit angle region ($\alpha_f < 0.3°$), multibeam scattering/interference fringes are prominent, matching the experimental data exceptionally well. The intensity oscillation along the $\alpha_f$ direction in the scattering profile, shown in Fig. 3d, indicates the surface pattern's out-of-plane form factor, namely its thickness. The relationship between the oscillation period of the intensity profile in Fig. 3d and the thickness of the sample can be significant because the sample's 3D structure can now be determined by a single scattering pattern at a single incident angle by using the FE-DWBA-based forward simulation. We note that the previous 3D reconstructions used the scattering data sets collected at multiple incident angles. An additional discussion of the out-of-plane dimension will follow in the next section. In Fig. 3e, it is visible that the fringe period along the $2\theta$-direction is sensitive to the pattern length along the X-ray beam direction (70 μm). The 3D structural determination shows the powerfulness of the FE-DWBA method if the scattering data is measured and analyzed properly. Therefore, the single-shot 3D information can be used to monitor time-resolved changes of samples under stimuli with a high spatial and possibly temporal resolution.

The FE-DWBA method is applicable to reveal the dynamical scattering of more complex samples, e.g., the petal-like planar pattern presented in Fig. 3f. Figure 3g shows the side-by-side comparison of the corresponding simulated and experimental scattering pattern. The fringes along $\alpha_f$ are even more prominent as the scattering data occupies more reciprocal space voxels due to the more complex 2D shape of the 3D sample. Again, the FE-DWBA simulation is well-validated by the experimental data. Because the FE-DWBA is based on layered cellular grids, it should be applicable to multilayer surface samples when a priori knowledge can be used to optimize the cell geometry and size in the direction normal to the surface. The advantage of the forward-calculation method is that it uses the full Parratt's formalism[31], which is well suited for computing dynamical scattering from surfaces and interfaces of complex planar 3D patterns such as multilayer and multicomponent structures in micro- and nanoelectronic circuits. However, the computation cost will be higher as fewer vertical stacks sharing the same electron density profile can be grouped. Although hard X-rays are less sensitive to chemical composition than soft X-rays and ultraviolet probes[65], the sub-nm wavelength promises higher spatial resolution or sensitivity, as shown in what follows next. Another practical issue with this bilayer sample configuration of 55-nm Au on top of the 5-nm Ti adhesion layer is that the simulation may only be sensitive to the nominal total thickness (60 nm), not the individual layer thicknesses. This is due to the grazing-incidence angle just slightly above the Au critical angle at which the relatively thick Au layer significantly attenuates the X-rays and the Au/Ti interface becomes less visible.

## Grazing-incident scattering holography
The successful FE-DWBA simulations demonstrate that the physics responsible for the fringes is similar to the waveguide effects[41]. A detailed analysis of the fringes reveals that an important phenomenon —Lloyd's mirror interferences, previously observed in other wavelength regimes, is now demonstrated in this hard X-ray grazing-incidence geometry near the critical angles. The periodic fringes seen in

the $\alpha_f$ direction in the scattering pattern correspond to a dimension of twice the sample's thickness. In SI #4 and Fig. S3, the scattering patterns taken at different incident angles (above the critical angle of the Si substrate) show that the period of the interference fringes is invariant for a given sample.

As illustrated in Fig. 4a, when a point at the 3D planar pattern surface or inside is illuminated by a coherent beam, the scattered beam becomes a secondary source. The sample scattering from the secondary source has two major paths, one directly leading to the detector (Path A) and the other reflected by the substrate (Path B). The scattered wave along Path B experiences an additional phase shift, a function of the exit angle. These coherent waves from the two paths superpose and create interference patterns observed at the detector location. We emphasize here that in the direction along the sample/substrate horizon (<0.3°), the waves on both Paths A and B are different from the incident (primary) X-ray beam and its reflection from the substrate. With an incidence angle (0.6°) above the critical angle, the former is attenuated by the substrate and would not reach the detector. The latter, the reflection of the primary beam, reaches the detector at a reflection angle the same as the incident angle (0.6° in this case). Therefore, the interference and holography patterns, providing the 3D sample information from the single view, are in the dark field, high contrast, and low noise, distinctly different from all previous Lloyd's mirror holography themes. At $\alpha_f = 0$, the Fresnel reflectance is close to unity, and the phase shift is $\pi$, resulting in a node in the interference intensity similar to X-ray standing waves at X-ray mirror surfaces[62,63]. More specifically, when one point at a distance $d$ above a mirror interferes with its mirrored point at $-d$, below the substrate surface, it behaves like a two-point source, separated by $2d$, giving rise to scattering fringes with a period of $2\pi/(2d)$. The characteristic thickness $2d$, directly determined by the interference period, is a dimension that doesn't exist in the sample. This indicates that the near-horizon fringes are generated from a scattering mechanism that is very different from the kinematic scattering theory, which shows a dependence only on the sample thickness. Figure 4b shows the pattern thickness calculated from Lloyd's mirror path difference is 57.2 nm. As $\alpha_f$ increases, the Fresnel reflectivity drops as $q_z^{-4}$. Smeared by diffuse scattering from the surface roughness, the interference intensity due to Lloyd's mirror effect decays and eventually is inundated in the background. Parenthetically, we note that the original Lloyd's mirror concept was deemed not practical for hard X-ray applications[46,66,67]. Now we demonstrate that it can be ideally suited for creating holography and coherent imaging at the surfaces as a high-resolution hard X-ray structural probe. The quantitative analysis of the dark-field Lloyd's-mirror-based interference and holography patterns is discussed next.

## The first-principles holography simulation
While the FE-DWBA method produces high-fidelity simulation results, like other numerical methods, the computation requires an excessive amount of computing resources. In many cases, the grid must be fine enough to match the experimental conditions, and the electric field in each cell must be computed. This may hinder its application in iterative image reconstruction algorithms, which involve multiple forward and backward calculations. Close examination of the four terms in Eq. S12 reveals that the first and the second term dominate if the incident angle $\alpha_i$ is above and the exit angle $\alpha_f$ is below the critical angles (i.e., the region where we observe the dynamical scattering fringes). The other two terms are relatively weak due to the scattering geometry. This enables the translation of the complex scattering process for samples on a surface into a much simpler approximation. The scattering is approximately the kinematically scattered wave of the sample mixed with its reflected wave from the substrate if the incident angle is above the system's critical angles, in a simplified scheme without the presence of the cells

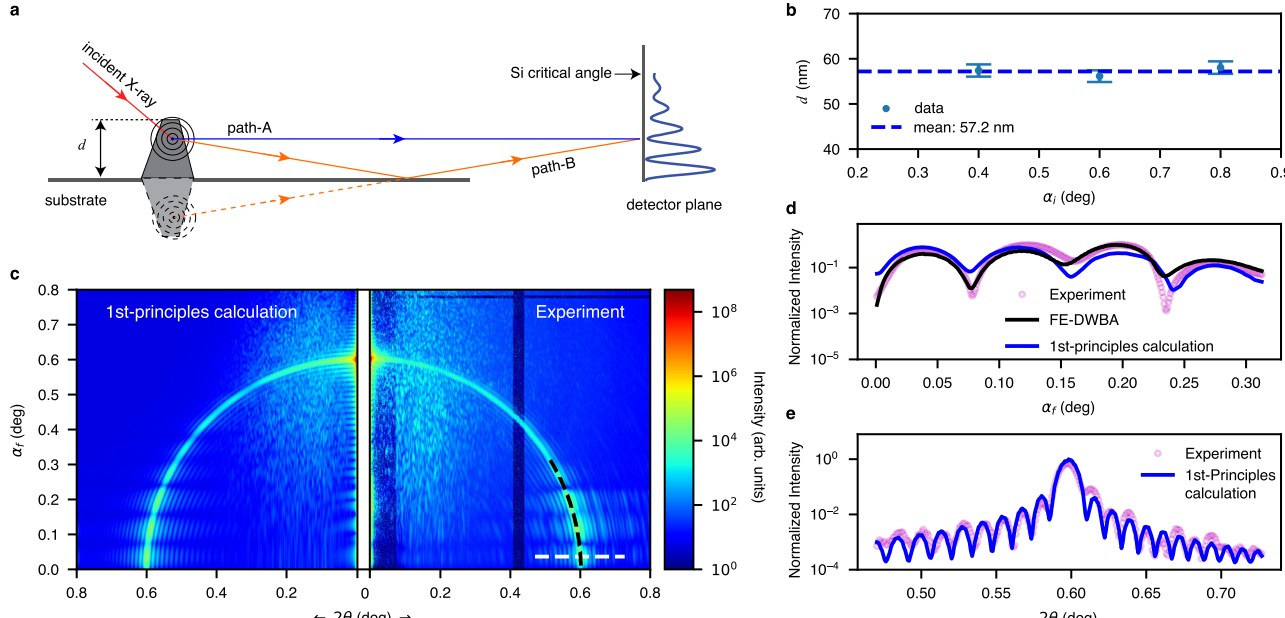

**Fig. 4 | Grazing-incidence X-ray holography of the substrate-supported single-bar surface pattern. a** Schematic of surface holography near the substrate (not to scale). The scattered X-rays from the sample at low exit angles (close to or below the critical angles) can propagate towards the detector via the two pathways—the direct Path A and reflected Path B. They have a path length difference of approximately $2d\sin(\alpha_f)$ and an incident angle-dependent reflection phase shift. When $\alpha_f$ is lower than or close to the substrate critical angle, the direct and reflected scattering intensities are comparable, creating a situation similar to Lloyd's mirror and Young's double-slit with an effective slit spacing of $2d$ and the additional variable phase shift due to the reflection. **b** The two-path interference period along the $\alpha_f$ direction is invariable for incident angles (0.4, 0.6, and 0.8°) above the substrate critical angle of 0.22°. The fringes from experimental images correspond to $2d$ of 114.4 nm (see text for details). The error bars are estimated from the uncertainty

(about 3 detector pixels) of the interference period along the $\alpha_f$-direction. **c** The holography-based first-principles calculation generates a simulated pattern (left panel) that well matches the experimental data (right). **d** Line scans along the center of the arc ($q_x = 0$) from $\alpha_f = 0°$ (horizon) to 0.3° (black dashed line in panel **c**), showing the first-principles Lloyd's mirror calculation is validated by both the experimental data and full-fledged FE-DWBA simulation at the low exit angles. **e** Line scan of the scattering profile along the white dashed line in panel **c**, containing the information of the form factor of thin-film surface pattern, more specifically, the length of the bar along the X-ray beam. The oscillation period indicates the bar length is 70.0 μm precisely. The first-principles calculation indicates that the single-incident angle scattering pattern contains the precise sample structural information in 3D.

and layers, the interfering scattered waves in Eq. 1 and S12 can be:

$$I\left(\alpha_i, \alpha_f, 2\theta\right) \propto \left| \int \rho(\mathbf{r})\left\{1 + R\left(\alpha_f\right)e^{-2ik_z(\alpha_f)z}\right\}e^{-i\mathbf{q}\mathbf{r}}d\mathbf{r} \right|^2 \quad (2)$$

where the first term is the unperturbed scattering (Path A in Fig. 4a) and the second is the mirror image of the first term but modified with a complex reflection coefficient, $R(\alpha_f)$, and the phase shift $2k_z(\alpha_f)z$ due to the mirroring effect by the total external reflection at the substrate surface (Path B, mirrored scattering, in Fig. 4a), in the condition of a higher incident angle ($\alpha_i$) and lower exit angle ($\alpha_f$). The distance between the two scattering sources at the sample location is effectively $2z$. With known substrate optical prosperities, the calculation of $R(\alpha_f)$ is relatively straightforward based on Parratt's algorithm, which also predicts $|R(\alpha_f = 0)| = 1$ with a phase shift of π upon reflection in Path B. This constraint is shown in the holography images as the scattering intensity $I(\alpha_i, \alpha_f = 0°, 2\theta) = 0$, so that the intensity at the detector corresponding to the direction of the sample/substrate horizon is always 0, an absolute constraint imposed onto the interfering waves, regardless of the incident angle and the sample structure. Therefore, within such angular ranges, the surface scattering is reduced to a two-beam holography mechanism, which can also be the preferred experimental condition for CSSI measurements. The simplified model demonstrates the true holographic nature of the multibeam scattering method that can be extended to using hard X-rays of tens-keV photon energy.

Based on the holography scheme, we developed a simulation using the simplified model in a first-principles approach that does not

require finite-element-based numerical computation, as detailed in SI #5. The simulated scattering with the simplified model again matches the experimental data very well, as shown in Fig. 4c. The line scans along the $\alpha_f$ (Fig. 4d) and $2\theta$ (Fig. 4e) directions also show an excellent match in both node and antinode positions, as well as oscillation periodicities related to the pattern thickness and the in-plane dimensions, respectively. The possibility of capturing all major scattering contributions validates our simplified computation method based on the two-beam holographic approach.

To better demonstrate the effect of the holographic process that reveals the accurate 3D form factors of the surface patterns, we designed a sample based on two stacked bars. The bottom bar has dimensions of 300 μm (length) × 6.0 μm (width, $w_b$) × 55 nm (thickness, $h_b$). Again, only the simulation is sensitive to the combined thickness of the Au and Ti layers for the same practical reasons mentioned previously. The top bar is also 300-μm long but 3.0-μm wide ($w_t$) and 30-nm thick ($h_t$). The true 3D sample is illustrated in Fig. 5a, including a 3D rendering, top view, and cross-section normal to the X-ray beam. The stacked bars create a non-uniform thickness and multiple interfaces in both y- and z-directions. In the sample fabrication process, the top bar was inadvertently misaligned by a minimal angle of γ = 0.009 (drawn not to scale). This angle was only revealed to be present by our first-principles calculation below. To our pleasant surprise, the misaligned bars offer several unambiguous tell-tale features in the experiment and simulation to demonstrate the sensitivity of the holographic scattering imaging to the nm-scale features in this truely 3D sample.

In the experiment, the X-ray beam impinged upon the left part of the sample with an incident angle of 0.6°, as the X-ray footprint is

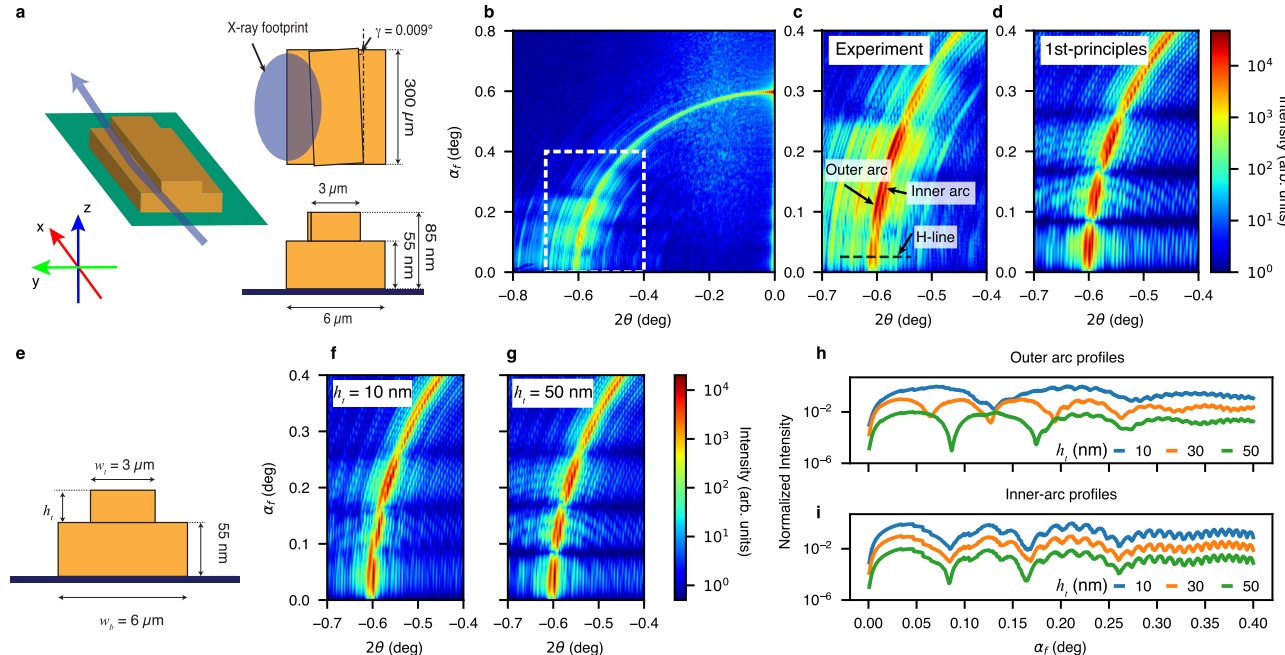

**Fig. 5 | Grazing-incidence holography of a stacked-bar pattern to demonstrate the true 3D nature of surface scattering: dependence on the pattern thickness.** **a** 3D schematics of the stacked-bar pattern deposited on a Si substrate with the X-ray incident direction indicated, top view, and side view along the X-ray direction with nominal (designed) dimensions (not to scale). The top view highlights an unintended misalignment between the long axes of the bottom and top bars of about 0.009°, introduced during the preparation of the top bar. **b** The left half of the CSSI pattern of the stacked bar was taken at an incident angle of 0.6°. The dual-beam holographic scattering features are outlined by the rectangular box at low exit angles $0 < \alpha_f < 0.4°$. At first glance, the pattern in the box is similar to the scattering from the single-bar pattern, but it has many additional fine features generated by the stacked bars. **c** Magnified view of the experimental data in the

dynamical multibeam scattering range of $2\theta = (-0.7 – 0.4°)$ and $\alpha_f = (0 – 0.4°)$. **d** First-principle simulation using the dual-beam holographic interference approximation. **e** Schematic of the stacked-bar sample pattern with fixed bottom-bar dimensions and a top bar with a fixed width of 3 μm but a varying height of $h_t$. **f**, **g** Simulated holographic scattering patterns in the white box region for varying top-bar thicknesses $h_t$ of 10 and 30 nm, respectively. **h** Simulated scattering intensity profiles (line scans) along the outer arc as a function of $\alpha_f$ from 0 to 0.4° for varying $h_t$ values 10, 30, and 50 nm. While the weaker and shorter-period oscillations remain similar over the full $h_t$ range, the large-period oscillations are sensitive to $h_t$ and vary dramatically. **i** Scattering profiles taken along the inner-ring arc showing that the profiles are insensitive to the top-bar height but sensitive to the bottom-bar thickness (see SI #7).

marked in Fig. 5a (top view), and generated a scattering pattern with rich features, especially in the low exit angle and dynamical scattering region (Fig. 5b). The most notable ones, highlighted in Fig. 5c, include two primary arcs (labeled as the inner and outer arcs) and multiple secondary but eccentric arcs around the primary ones. The origin of the two-arc scattering pattern is discussed in SI #6 and Fig. S4. Using the two-beam holographic computation approach, we were able to model the stacked bars with seven parameters (i.e., dimensions of the two bars of identical length, the relative location of the top bar, and the misalignment angle), and simulate a scattering pattern that contains all the major features present in the experimental data, as shown in Fig. 5d. The simulation clearly shows that the double-arc results from the tiny angular misalignment of the top bar with respect to the bottom one. Although a perfect match of the entire speckle patterns between the experiment data and the simulation is not realistic, especially using a scattering model with significant approximations, the level of the quantitative agreement is remarkable, as demonstrated as follows. The complexity of the scattering pattern/speckles, even at a single fixed incident angle, is apparent, and all the global and fine scattering signatures are contributed from all dimensions of the entire 3D sample pattern and the fine features in 3D.

To illustrate the holographic imaging concept and explain how the sample's 3D dimensions impact the scattering patterns quantitatively, we first demonstrate explicitly that the sample information in the third dimension (i.e., normal to the stacking direction) is encoded in a single-view scattering pattern. For clarity, we use the top-bar height as the only variable, as shown schematically in Fig. 5e, and we performed simulations for three top-bar heights, $h_t$, of 10, 30, and

50 nm. Figure 5f, g show the simulations with $h_t = 10$ and 30 nm, respectively, to highlight differences. The change of the large-period modulation along the outer arc is immediately apparent. The oscillation period is susceptible to $h_t$ and varies dramatically over the range. In Fig. 5h, the scattering profiles were taken along the outer arc as a function of $\alpha_f$, showing the impact of the top-bar height unambiguously. In the meantime, the corresponding oscillations on the inner arc (Fig. 5i), more sensitive to the bottom-bar dimensions, specifically the thickness, remained invariant. More details are presented in SI #7 for the variation of the bottom-bar thickness $h_b$. Therefore, the scattering intensity on the outer arc is mainly contributed from the top bar, while the inner arc is from the bottom bar.

To understand the correlation between the intensity oscillation at the higher frequency and the sample dimension, we vary the width of the top bar, $w_t$, (see Fig. 6a) from 2.5 to 4.0 μm, and two simulation results are shown in Fig. 6b, c. Note that the calculation of the scattering pattern in Fig. 5d uses the ground truth $w_t = 3.0$ μm. In addition, we observe shorter-period (higher frequency) intensity oscillations superimposed on both inner and outer arcs along the $2\theta$ direction. They remain similar and independent of the $h_t$ and $w_t$ variation. In Fig. 6b, c, the effect of the top bar width is shown clearly in the 2D patterns. Quantitatively, in Fig. 6d, the integrated intensity profiles of the inner arc along $\alpha_f$ as a function of $2\theta$ contain many short-period high-contrast oscillation corresponding to the distance between the left edges of the top and bottom bars. The oscillations also appear in the profile of the outer arc (Fig. 6e) but with a much-lessened contrast. As a result, the inner-arc profiles show high contrast in the fringes, whose periodicity is inversely proportional to $(w_b - w_t)/2$, as the

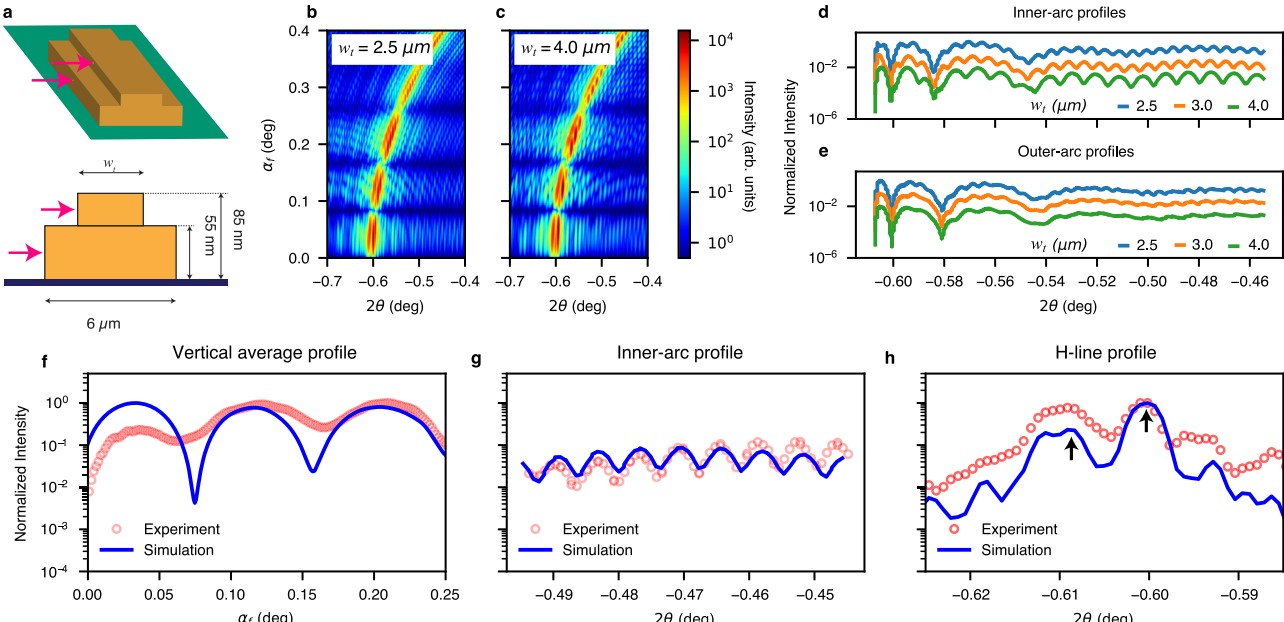

**Fig. 6 | Grazing-incidence holography dependence on the stacked-bar pattern width and quantitative agreement between the data and holographic simulation. a** Schematic of the stacked-bar sample pattern with fixed bottom-bar dimensions and a top bar with a fixed thickness of 30 nm but a varying width $w_t$: 3D rendering and a cross-sectional view along the X-ray beam incidence. **b, c** Simulated holographic scattering patterns for $w_t = 2.5$ and $4.0\,\mu m$. **d** Scattering profile taken along the inner-ring arc as a function of $2\theta$ from $-0.60$ to $-0.45°$ showing the short-period oscillations are strong and the period is inversely proportional to the distance between the side surfaces (marked by the two red arrows in panel **a**) in y-direction or $(6\,\mu m - w_t)/2$, as $w_t$ varies from 2.5 to 4 $\mu m$. **e** Simulated scattering intensity profile along the outer arc as a function of $2\theta$ as $w_t$ varies from 2.5 to

4.0 $\mu m$. While the larger-period intensity profile remains unchanged over the $w_t$ range, the short-period modulation is observable but weak. **f** Scattering intensity profile, integrated over a narrow range of $2\theta$ as a function of $\alpha_f$ shows the best agreement between the simulation and experiment data when the bottom-bar thickness is 55 nm. **g** With the sensitivity of the inner-arc scattering intensity profile to the top-bar width $w_t$, the best match between the experimental data and the simulation reveals $w_t = 3.0\,\mu m$. **h** The first-principles holographic simulation shows the sensitivity of the angular displacement between the outer and inner arcs to the top and bottom bars' axis rotational misalignment. The simulation reveals that the top bar's long axis has an unintended misalignment of 0.009° with respect to the bottom one, counterclockwise in the top view.

bottom-bar width $w_b = 6\,\mu m$. The in-plane dimension change introduces a fine but clear impact on the simulated pattern—on both inner and outer arcs, the signature of the interfering scattering off the vertical sides of the 3D structure at the different heights and horizontal surfaces of the top and bottom bars.

The first-principles two-beam interference calculation demonstrates quantitative agreement with the experimental data. Figure 6f shows the scattering intensity profile, integrated over the $2\theta$-range of $\sqrt{0.6^2 - \alpha_f^2} \pm 0.03$ (°) as a function of $\alpha_f$, showing the dynamical scattering in the low-exit angle region. Since the X-ray beam footprint is off-center near the edge of the sample, the interference oscillation of the integrated intensity profile is dominated by the thickness of the larger bottom bar, 55 nm. The high-frequency inner-arc profile confirmed the width of the top bar of 3 $\mu m$ by the excellent agreement between the simulation and the experiment (Fig. 6g). The first-principles holographic simulation is sensitive to the angular displacement between the outer and inner arcs, shown in Fig. 6h. The simulation indicates that the top bar's long axis has an angle of 0.009° with respect to the bottom one, counterclockwise in the top view. This slight angle originates from the misalignment during the preparation of the top bar, which was due to hardware limitations of the lithography instrument. Surprisingly, it illustrates the extreme sensitivity of the surface holography method when probing elongated structures, such as micro- and nanoelectronics circuits and masks. With the sensitivity of the scattering patterns in the lower exit angle region to the width and height of the bars, every sample dimension can be obtained by fitting the experimental data and the simulations, which agrees well with the designed parameters.

The sensitivity of this multiple scattering probe is further investigated as a function of the tell-tale features, again using

examples such as the top layer width ($w_t$) and thickness ($h_t$). In principle, the sensitivity (or spatial resolution) is determined by the scattering angles (or maximum momentum transfer in the respective directions). We estimated that in the $2\theta$ direction, the value is limited by the detector sensor edge at 0.7°, which resulted in a resolution of *ca.* 5 nm. A direct demonstration is shown using numerical simulations with varying top bar width by a few nm around a hypothetic 5.000 $\mu m$. We chose this value different from the actual sample dimension (3 $\mu m$) to avoid reduced contrast in the interference fringes because of the detector pixel size effect in the experimental data, as shown in Fig. 6d. In Fig. 7a, we illustrated the line scans of the simulated scattering patterns with variations ($\Delta w_t$) of 3-nm step size. While there are detectable differences in the speckle patterns, the varying speckle sizes (or interference periods) are immediately observable in the line-scan plots, Fig. 7a, showing the striking effects of 3-nm changes of the form factor. The finer sensitivity value, better than 5-nm afforded by the detector data range, is from the over-sampling of the interference fringes. Therefore, we think that the probe sensitivity or resolution can be even better than 3 nm, especially when higher coherent X-ray flux, coupled with larger-format detectors, is used to generate the scattering images.

In the out-of-the-plane or z-direction, the detector samples an $\alpha_f$ range (or momentum transfer) much larger than the y-direction, which implies better sensitivity. A similar simulation is performed with a variation ($\Delta h_t$) and finer step of 1 nm around the top-bar thickness ($h_t$) of the actual 30-nm value, shown in Fig. 7b. Again, the line scans demonstrate 1 nm, if not sub-nm, sensitivity in measuring the top-bar thickness. The simulation results of $\Delta w_t$ and $\Delta h_t$ resemble an interferometer when the shift of fringes is sensitive to small changes in optical paths.

## Discussion

In a most direct fashion, we demonstrated that coherent surface scattering at grazing incidence contains strong multiple scattering components, which encode the true 3D morphology of surface structures. More importantly, this 3D information can be extracted from a scattering pattern taken at a single incident angle, which does not require scanning the incident angle as required by the conventional CSSI analysis and should be proven suited for obtaining critical-dimension information in mesoscale surface patterns. To take advantage of this capability, we developed the FE-DWBA method that captures the origin of multiple scattering physics and reproduces scattering patterns with high fidelity. Furthermore, aided by the FE-DWBA simulation, we discovered that the multiple scattering resembles Lloyd's mirror in the hard X-ray regime revealing the two-beam holography concept in the grazing-incidence condition. In the holography framework, a much-simplified first-principles scattering simulation reduces the computational complexity even further while capturing the detailed 3D surface morphology qualitatively and quantitatively. The simplification enables parametrizing the real-space object without the spatial resolution limit imposed by FE-DWBA and other finite-element-based scattering simulations, paving the way for applications such as 3D X-ray metrology of mesoscopic surface patterns with the improved spatiotemporal resolution or sensitivity of a few nm (in-plane and transverse to the X-ray beam) and even sub-nm in the out-of-plane direction. In addition, the multibeam scattering simulation and reconstruction have a significant bearing on electron-beam and extreme-ultraviolet (EUV) imaging and scattering, where multiple scattering is a more commonly occurring phenomenon due to the strong interaction between the probes and matter.

The FE-DWBA simulation focused on one of the simplest samples to illustrate the principle of the analysis, the tell-tale features of the coherent scattering patterns, and the sensitivity to the pattern's 3D structures in a single incident-angle view. The simulation was also validated by the scattering images from a more complex pattern. Even though FE-DWBA treats the in-plane electric-field boundary conditions with approximations, the simulation results are of high fidelity with minimal computation load and are validated by the experiment data. We note that computational electromagnetics techniques, such as the finite-difference time-domain method[68], can solve the distribution and propagation of the electromagnetic wave more rigorously. Still, they require enormous computational resources for X-rays with sub-nm wavelengths, even for simple problems with low spatial resolution.

The simulation and experiments in this work are based on simple substrate-supported patterns with single or double layers of a single material. The FE-DWBA takes layered approaches in each cell, which resembles the X-ray reflectivity algorithm[31] and is best suited for simulating multilayer samples of different compositions. However, the sensitivity to the chemical compositions has yet to be tested in the current simulation, which is the subject of future work. It is recognized that the chemical sensitivity of hard X-rays would be inferior to that in soft X-ray or EUV regimes[65,69]; anomalous scattering may improve the sensitivity if one of the constituent elements is tracked during structural transformation.

The concept of X-ray holography with Lloyd's mirror setup was previously developed to use the primary and its reflection beams as the reference waves, which requires the X-ray source to be coherent and divergent so that the reference beam can superpose with the reflected beam. The two conditions are difficult to satisfy simultaneously, especially in the hard X-ray regime. In addition, the contrast and signal-to-noise ratio in such a bright-field hologram is also limited because the reference beam is usually much stronger than the reflected beam. The surface holography technique developed in this work uses the sample scattering as the secondary source and its mirrored image (with well-characterized exit-angle-dependent intensity reduction and

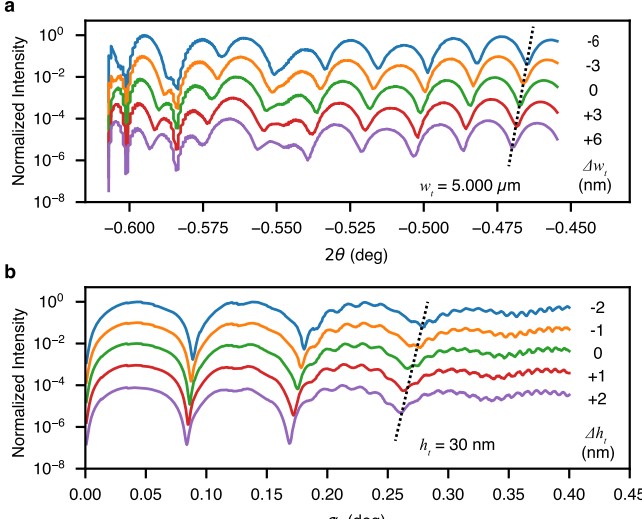

**Fig. 7 | Spatial sensitivity of the speckle size and phase of the holographic scattering on fine changes in the critical dimensions of the stacked-bar sample. a** Line scans of the inner-arc intensity as a function of the in-plane scattering angle $2\theta$ at various fine deviations $\Delta w_t$ in top-bar width $w_t$. **b** Line scans of the outer-arc intensity as a function of the off-plane scattering exit angle $\alpha_f$ at various fine deviations ($\Delta h_t$) in the top-bar thickness $h_t$. The black dashed lines guide the eye and highlight the speckle phase shift.

phase shift) as the reference beam. Thus, similar to dark-field imaging, all interfering waves from the sample are well separated from the direct beam and its specular reflection. With the use of high electron-density substrates (such as noble metal coated silicon or glass), highly coherent flux X-ray sources, and single-photon sensitive detectors, it's expected that the surface hologram in the grazing-exit-angle direction will increase significantly, unlocking new possibilities of surface characterizing technologies and application.

The simulation methods and algorithm are limited to the forward calculation for illustrating multibeam coherent surface scattering and holography and have yet to be extended to reconstruct the real-space structure. As we observed, the information on the coherent scattering is abundant when the exit angle ($a_f$) is between 0 (sample horizon) and the nominal critical angles of the sample, which should not have existed in the kinematic approximation. We demonstrated in this work that this multibeam scattering region contains rich and crucial information about the sample form factors in all three dimensions. It will be our future work to evaluate whether the patterns in this 'forbidden' area have enough speckle oversampling rate for a full-fledged reconstruction. In principle, with the forward-calculation algorithms such as FE-DWBA and the first-principles analysis, reconstruction using the scattering images with a priori information should be possible in the multibeam scattering regime. However, the feasibility of such reconstruction on the complex 3D subject remains to be determined. An algorithm using the DWBA approach has been attempted but is yet to be validated experimentally in the case of a weak perturbation to the in-plane electric field[44]. With strong sample-substrate multiple scattering, such as the system studied here, one of the solutions is to use the conventional kinematic reconstruction as a support to facilitate high-fidelity dynamical reconstruction. Conversely, the CSSI reconstruction can also benefit from the 3D morphological information captured in single-view holographic images.

With the advent of novel fourth-generation X-ray sources with more coherent flux, holographic imaging with dynamical surface scattering will immediately benefit the research communities interested in diffractive imaging, surface characterization, X-ray metrology, and X-ray reflective optics design. The 3D characterization capability

associated with the surface coherent scattering method is set to answer several scientific questions more effectively than conventional coherent imaging techniques. For example, how metrology of 3D micro- and nanoscale electronics is performed nondestructively[1], how thin films and quantum dots grow at surfaces and interfaces[3], how to advance nanopatterning using a combination of top-down and bottom-up techniques for the controlled fabrication of complex and multicomponent nanomaterials, how to control the morphology of thin-film photovoltaics to optimize the efficiency of the devices[70], and how physical and chemical processes and dynamics lead to hierarchical order in mesoscopic structures and the functionalities[2,71]. With the surface enhancement due to the multibeam scattering, especially waveguide enhancement[39,40,43], it should provide the best opportunity to study the structures and dynamics of biological membranes and supramolecules evolving in aqueous environments[6–8].

One of the unique capabilities stemming from the dynamical or holographic scattering is the single-view 3D analysis of a surface pattern's structure. This will prove useful for time-resolved in situ and operando measurements in realistic sample environments. We note that 3D structure determination from a single view has been long sought to benefit broad fields of physical and life sciences[72], which may now be enabled by the surface holographic imaging analysis developed in this work.

## Methods

### Samples

The samples studied in the present work were prepared at the Center for Nanoscale Materials, Argonne National Laboratory (ANL), using electron-beam lithography (EBL) and electron-beam evaporation techniques. The substrates used were 3-mm-thick flat silicon wafers. A poly(methyl methacrylate) (PMMA) resist layer was spin-coated onto the silicon wafer and dried using a heating plate. The sample was patterned with a design incorporating alignment marks for multi-level samples using a JEOL JBX-8100FS EBL system. The exposed regions were washed away using a mixture of water and 2-propanol (3:7 ratio by volume). A 5-nm Ti layer was deposited onto the silicon substrate as an adhesive layer to bind the silicon substrate and the first gold (Au) layers (50 nm) of the micro-patterns. Both layers are deposited using a Temescal FC2000 E-beam evaporator. The stacked-bar sample required a second exposure and electron-beam deposition. After the bottom bar was fabricated, the sample was coated with the same PMMA resist and aligned using the marks to determine the position for the second EBL exposure. A nominal 30-nm thick Au layer was deposited directly on the bottom Au layer without any adhesive layer. The top and bottom bars were intended to parallel each other as the top bar was to situate at the center of the bottom one. Nevertheless, a slight misalignment of 0.009° was discovered later in the measurement and analysis. Although the alignment process for the second EBL step included multiple points to correct for angular error, small (nm length) drifts can occur during the EBL patterning, creating linear and angular misalignments.

**Coherent surface scattering imaging experiments.** The CSSI experiments of the single-bar sample were performed at P10, a dedicated coherent scattering beamline at the PETRA III synchrotron facility at DESY, using an X-ray photon energy of 8.0 keV (wavelength $\lambda = 0.155$ nm). The monochromatic and coherent X-ray beam was focused to $1.7 \times 3.5$ (V × H, FWHM) $\mu m^2$ using a set of compound refractive lenses (CRLs). The detector was an EIGER X 4M with $75 \times 75$ $\mu m^2$ pixels placed 5.0 m from the sample with a vacuum flight path in between. Scattering patterns collected at slightly offset positions were superimposed to fill the gaps between the detector sensor modules. The coherent images were collected at different grazing-incident angles from 0.2 to 0.8°. The coherent flux at the sample location was about $0.5$–$1.0 \times 10^{11}$ photon/s, and detector exposure time

for collecting a coherent scattering pattern was between 1 and 64 s depending on the incident angle.

The stacked-bar sample was measured at the 8-ID-I beamline of the Advanced Photon Source, ANL. A set of CRLs focused a 7.35-keV monochromatic X-ray beam to a $2.2 \times 2.5$ (V × H, FWHM) $\mu m^2$ spot size at the sample location. A Lambda 2 M detector with $55 \times 55$ $\mu m^2$ pixels was placed at 3.95 m downstream from the sample with a vacuum beam path in between to reduce air attenuation scattering. The coherent X-ray flux was about $5 \times 10^9$ photon/s, and the exposure time was 0.5 s. The images were taken at a fixed incident angle of 0.6°.

## Data availability

The experimental and simulation data generated in this study have been deposited in the Figshare database [https://doi.org/10.6084/m9.figshare.23401400.v2].

## Code availability

The simulation and computational codes for this study are available from the corresponding authors upon request.

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

## Acknowledgements

We acknowledge DESY (Hamburg, Germany), a member of the Helmholtz Association HGF, for providing experimental facilities. Parts of this research were carried out at PETRA III, and we would like to thank Fabian Westermeier and Dmitry Dzhigaev for their assistance at the P10 beamline. Beamtime was allocated for proposal I-20170155. We acknowledge the fruitful discussion with Sunil K. Sinha. We also thank Wei Jiang for our early discussion on the algorithm and Suresh Narayanan and Raymond Ziegler for their valuable assistance with the experiments at Sector 8 of the Advanced Photon Source (APS). We acknowledged that Jong Woo Kim and Pice Chen who contributed to the sample preparation and the data collection and Donald Walko who read and commented on part of the manuscript. Work was also performed at the APS Center for Nanoscale Materials of Argonne National Laboratory (ANL), both US Department of Energy (DOE) Office of Science (SC) User Facilities. J.W., Z.J., M.C., M.W., and T.S. acknowledge the funding support provided by Laboratory Directed Research and Development (LDRD, 2017-073-N0) from ANL and by the US DOE, SC, Office of Basic Energy Sciences (BES), under Contract No. DE-AC02-06CH11357. J.W., Z.J., and M.C. acknowledge partial funding support from the Accelerator and Detector Research Program of the US DOE, SC, BES under Contract No. DE-AC02-06CH11357. Z.J. acknowledges partial support from the US DOE Early Career Research Award.

## Author contributions

J.W., M.C., and Z.J. conceptualized the experiment and the study. M.C., Z.J., T.S., M.S., and J.W. performed experiments. M.W. and M.C. prepared the samples. M.C., J.W., and Z.J. interpreted the data. M.C. (assisted by Z.J. and J.W.) developed the analysis method and algorithms and performed the simulations. M.C. prepared the first draft. All authors contributed to revising the manuscript. J.W. and M.C. wrote the final version of the manuscript.

## Competing interests

The authors declare no competing interests.
