## [Peer Review File · Nature Communications]

Probing three-dimensional mesoscopic interfacial structures in a single view using multibeam x-ray coherent surface scattering and holography imagingREVIEWER COMMENTS

Reviewer #1 (Remarks to the Author):

Miaoqi et al presents an interesting set of grazing incidence coherent X-ray scattering experiment that is complemented by a detailed quantitative analysis. The paper shows beautifully that in an X-ray scattering experiment, even if the experiment is simple, detailed analysis can reveal nanoscale results that are not that easy to get from other techniques. Having said that, the article is difficult to follow and at some times gives the impression that it is written for an audience that is pretty well versed with advanced X-ray methods. The results of the main article cannot be understood without a substantial reference to the SI. In my opinion the work presents a novel method and significant advancement, and should be published.

Below I put in my comments/suggestions:

The main article is quite long and repetitive. I suggest to write in a way that is easy to navigate. As of now, going back and forth between the main article and SI is hindering the readability and understanding. Even the Introduction seemed too long. On top the combination of two relatively difficult topics of surface scattering and X-ray holography complicated readability. Near the middle of the article the topic of holography is introduced which seems almost a by-product of the scattering analysis. In fact once I went through the holography section, I felt it undermined the FE-DWBA section.

A lot of work has been done by the imaging community, CDI, Ptychography, Bragg ptychography, time dependent imaging etc. I did not find any reference to them. In fact, in my view a paragraph putting the current work in context to the imaging work will be important. There are many instances where it is implied that CDI cannot work in case of multiple scattering (line 8). What is difficult to understand is that, if a real space image with nanometer resolution (at least tens of nm) is obtained, then that solves the structure problem. Or, are the authors claiming that the imaging community had not obtained the right image? Asked in a different way, what new structure information the current technique provide that cannot be obtained using CDI/Ptycho?

An explanation of the origin of the intensity pattern will be very helpful. As it is, the paper quickly goes into analysis, but what general information is obtained from the complicated intensity pattern? After all, it is an experiment work, and as a result a clear explanation/understanding of data is important.

The FE-DWBA is an interesting concept. Slicing of a thin film into several layers to fit reflectivity curves is well known. In this work the authors have applied similar idea but in all dimensions. By replacing height-height correlation by creating a grid, each one with a different refraction index, while maintaining the boundary conditions. A detailed explanation in the main text will be helpful. How is the initialization done? What are the constraints and how are they applied? Can the method work for a heterogeneous thin film? What is the computation cost of the method?

In the holography section, the first comment that I have is, as before, I found missing context. No referencing of holography or X-ray holography has been done. As mentioned above, both in surface scattering section and holography section, the referencing needs to improve. It should include current work done in hard and soft X-rays to inform readers of the state of the art in the field.

A hologram is a method whereby signal is mixed with a reference wave (usually plane wave) so that phase information is coded in. Thus, the beauty and strength of a hologram is in obtaining real space image. As best as I can understand, that is not the case here. In the present case, it looks like a two

beam interference effect. Further, B is a point on the substrate? Shouldn't the scatter beam be significantly modified to be a reference beam to create a hologram. Why doesn't it create its own speckle pattern? From the data it looks like two co-propagating beam suffering similar scattering/reflection mechanism.

Is it possible that at such low angle the whole sample is becoming an effective point scatterer due to "shortening" of the sample. Thus these fringes at long alpha but high 2theta are basically pseudo-Airy fringe type scattering?

Will the method not be useful in case there is no multiple scattering? That makes the use-case very limited. A comment would be helpful.

Reviewer #2 (Remarks to the Author):

This manuscript reports on the development of a technique that uses the scattering of a doubly focused coherent X-ray beam under grazing incidence to characterize micro-nano-structures with large dimensions, of the order of those of the coherent beam. They show that, in this geometry, 2D images of the coherently scattered beam are very sensitive to every details of all the morphological features (shape and dimensions). They develop an approach to simulate the data, which is based on a combination of the well-known Distorted Wave Born Approximation (DWBA) and Parrat matrix used formalism to analyze X-Ray reflectivity. This approach, although based on a first order approximation, is shown to account for every details on the morphology. This new approach is based on a Finite Element description of the sample, and a propagation, reflection and refraction of the X-ray waves between neighboring elements. With this so-called FE-DWBA description of coherent surface scattering, they demonstrate that each feature of the sample morphology yield characteristic features in the image. They remark that the images feature related to the sample height have a width twice smaller than expected for standard single scattering, and demonstrate that this is related to a holographic effect under grazing incidence in which coherent scattering between the object and its image with respect to the substrate plane interfere coherently. This is the so-called Lloyd's effect, which is well-known in the wavelengths range of radio waves, lasers and soft x-rays. They deduce that all the morphological information can be inferred from a single measurement a single grazing incidence angle larger than the critical angle for total external reflection, as opposed to more conventional coherent GISAXS (or coherent surface scattering), which requires measurements taken at different incident angles to properly analyze out-of plane features. The authors further demonstrate the extreme sensitivity of the technique to tiny variations of the morphology, on an object having a very small disorientation between two of its elements. By further neglecting the two minor scattering terms of the DWBA, the authors propose a much simplified version of the FE-DWBA analysis, which, like for GISAXS, corresponds approximately, if the incident angle is larger than the system's critical angles, to the mixing of the direct kinematically scattered wave and the wave scattered after its reflection by the substrate. Such a simulation does not require finite element-based numerical computation, and is thus much faster than the full FE-DWBA simulation.

They conclude that, with its 3D characterization capabilities, this new surface scattering method opens a wide field of characterization studies of objects supported by surfaces in a more effective way compared to other conventional coherent imaging techniques. Example of potential future studies include examples the metrology of 3D micro and nano -objects, the controlled fabrication of complex and multicomponent nanomaterials; the morphological characterization of thin-films photovoltaics or bio-membranes and supra-molecules in their aqueous environments etc.. Finally, they predict that, especially with the advance of novel 4th generation x-ray sources, such dynamic or holographic scattering will prove extremely useful for time-resolved in situ, in operando and possibly real-time measurements in realistic sample environments.

This work establishes a new step forward in the capabilities of X-ray scattering, using a small focused,

coherent beam under grazing incidence, to characterize a large diversity of systems, nondestructively, in very close to real conditions. They not only demonstrate the high potential of the technique, but also develop two new ways to analyze the data without resorting to full, extremely lengthy, full-fledged computational electromagnetics (CEM) techniques. This work is thus novel, and is based on further elaboration of previous works in the fields during the last decades.

The work is detailed, very well presented, supported by detailed calculations and computational work, and supports most of its conclusions and claims. Its capability to fully characterize the complex morphology of a single object made of a single material with high density on a low-density substrate is clearly established. However, I would appreciate some discussion about its possible limitations in characterizing less straightforward objects such as multilayer, multicomponent, nano-patterned micro-nano electronics containing very diverse elements of different densities. I would also appreciate some discussion on the accuracy to which the sample dimensions could be deduced and how fluctuations or roughness of the edges of or interfaces within the sample could be taken into account. The data interpretation, analysis and analysis development and discussion are built on firm grounds and are perfectly accurate. The conclusions are also strong and the work is shown to open a very wide field of studies. I would appreciate some more discussion and elaboration not only on the potential but also on the limits of the techniques, as just discussed above. The methodology is perfectly sounds and the work meets very high standards. The methods are provided with enough details for the work to be reproduced.

Reviewer #3 (Remarks to the Author):

This paper presents two related ideas for simulating the scattering patterns in coherent surface scattering experiments. First, the authors introduced what they call FE-DWBA (finite-element distorted wave Born approximation) to address the coherence of the beam in coherent surface scattering measurements. Conventional DWBA does not work well when the sample length scales are about the size of the coherent beam. Secondly, based on their FE-DWBA results, they introduced a simpler version referred as the 'first-principle holography simulation' that reduces the complexity of the simulations to a situation more akin to Lloyds mirror interferometer of only 2 beams mixing.

The authors are highly commended for this work. Their work adds significant insight to the measured complex scattering patterns and is a notable contribution to this field. This work invites further research to refine and extend the general technique of coherent surface scattering.

However, a primary criticism this reviewer has is that in many places, the paper conflates techniques that can truly reconstruct a 3D image (eg, CSSI in ref. 22, or CDI or ptychography) with what they have done, which is the ability to calculate scattering patterns of a-priori known samples that resembles the measurements. An accurate statement about their discovery is the first sentence in the Conclusion and Discussion: 'In a most direct fashion, we demonstrated that coherent surface scattering in grazing incidence contains strong multiple scattering components, which encode the true 3D morphology of surface structures'. In other words, there is 3D information embedded in the measured scattering patterns. This is perfectly fine. But it does not imply, nor have they demonstrated, the ability to reconstruct the 3D sample, just from the measured scattering patterns. There are several statements in the text (see below) that suggests the ability to determine the 3D structure from a single coherent surface scattering pattern.

The basic physics of this technique is that the substrate surface reflects a subset of the sample-scattered beam resulting in interference patterns which the authors have done an excellent job of simulating. However, the ability of this technique to provide true 3D morphology in a single-view has not been demonstrated and would seem to be extremely difficult without a-priori knowledge. Due to the geometry and substrate critical angle, each measurement only captures a subset of the sample

scattering. For example, since the incident beam is $>$ critical angles, the near $q=0$ scattering will remain $>$ critical angle and so, will not be encoded in the interference patterns. That said, the technique could be useful in determining small 'in-operando' changes in morphology if the 'original' 3D structure of the sample was known a-priori.

Some questions:

In the top 2 figures of Fig 2b are the wave-component assignments (TfTf and TfRf) flipped? If so, the same occurs in Fig S3.

Figure S3 caption has a reference to 'Eq. (x)'

Figure S2 caption first sentence suggests that simulations of BA and DWBA are shown, but in fact only the BA simulations are shown in Fig. S2. The DWBA simulations are shown in Fig. S3.

What causes the discrepancy between the simulations and experiment measurement near $2\theta=0$ in Fig 2c and Fig 3c? The simulations show strong fringes near $2\theta=0$ along the α direction but not seen in the measurement. Fig S4 explains the broad fringes for $\alpha < 0.4$ deg, but does not say anything about the finer fringes between $0.6 \text{ deg} > \alpha > 0.4 \text{ deg}$.

In the discussion on applications, the authors mention biological membranes and supra-molecules in aqueous environments. Seems like a very long stretch. Membranes scatter very weakly and tend to have unknown complex folds.

Given this improved 'forward modeling', have the authors tried to implement them (with multiple views) into the iterative reconstruction algorithms used in ref [22] ? Perhaps it would lead to improved reconstructions?

Examples of not quite accurate statements:

Abstract, first sentence: 'Visualizing surface supported and buried planar mesoscale structures such as nanoelectronics, ultrathin-film quantum dots, photovoltaics, and heterogeneous catalysts, require high-resolution imaging with coherent surface x-ray scattering'. This statement is clearly not accurate – imaging of nanoelectronics has been performed using ptychography and transmission x-ray microscopy, both of which are not 'coherent surface x-ray scattering'.

Abstract, second sentence: 'Here, we discover that the dynamical or multibeam scattering in grazing-incident reflection geometry promises three-dimensional structural determination in a single view but cannot be reconstructed by the conventional Fourier-transform approaches'. As mentioned above, this sentence is quite a stretch.

Abstract, last sentence: 'The holographic imaging paves the way for single-shot 3D structure determination...'. As discussed above, this sentence is not accurate.

Page 10: 'The relationship between the oscillationbecause the sample's 3D structure can now be determined by a single scattering pattern at a single incident angle.' As discussed above, this sentence is not accurate.

In two places (eg, second to last sentence in Abstract, and second to last sentence in Introduction), the authors mention 'nanometer resolutions' and 'sub-nm spatial resolution' respectively. They suggest that the scattering patterns can reveal such resolutions. There is no data to support these claims. The simulations in Fig 4 (h,i,m,n) do not support such claims.

In summary, I would recommend this paper for publication in Nature Communications after a careful

and substantive text edit. The work is technically sound and appropriate for Nature Communications, but the text needs to be improved. This is a highly commendable technical piece of work and it deserves to be published and read – in a clear and un-inflated narrative.

Reviewer 1:

Miaoqi et al presents an interesting set of grazing incidence coherent X-ray scattering experiment that is complemented by a detailed quantitative analysis. The paper shows beautifully that in an X-ray scattering experiment, even if the experiment is simple, detailed analysis can reveal nanoscale results that are not that easy to get from other techniques. Having said that, the article is difficult to follow and at some times gives the impression that it is written for an audience that is pretty well versed with advanced X-ray methods. The results of the main article cannot be understood without a substantial reference to the SI. In my opinion the work presents a novel method and significant advancement, and should be published.

We appreciate the reviewer's recognition of our experiments, quantitative analyses, and results. We also thank the reviewer for the criticism and comments that are valuable to us for improving the presentation of the manuscript. The comments are addressed in this point-by-point response. Overall, to orientate non-expert readers, we added a few sentences describing the focus of the analyses and the uniqueness of our methods of treating the coherent scattering data in terms of the incident and scattering angles rather than the 2D or 3D momentum transfers in conventional coherent scattering reconstruction (Pages 8 and 9). We also moved some SI materials to the main text, for example, the demonstration of the conventional DWBA analysis that fails to match the data satisfactorily, which inspired the new analysis presented here (Pages 9 and 10, and the new Figure 2). Also, we added the essential description of the FE-DWBA in the text (Pages 11, 12). We hope the reviewer now finds the reorganized text more reader-friendly than the previous version.

The main article is quite long and repetitive. I suggest to write in a way that is easy to navigate. As of now, going back and forth between the main article and SI is hindering the readability and understanding. Even the Introduction seemed too long. On top the combination of two relatively difficult topics of surface scattering and X-ray holography complicated readability. Near the middle of the article the topic of holography is introduced which seems almost a by-product of the scattering analysis. In fact once I went through the holography section, I felt it undermined the FE-DWBA section.

We agree with the reviewer that we are dealing with two sophisticated topics in the article. While writing up the manuscript, we considered presenting them in two related but separate manuscripts. However, since we used similar sets of samples and experiment data in the two different analyses, putting both topics in a single article would help to disseminate the ideas and cross-examine the simulations, which resulted in a relatively long main text (including the introduction) with a substantial SI document. Therefore, given the nature of this article, it would be difficult to eliminate the need to go back and forth between the main article and SI. To address the reviewer's comments, we made the following revisions:

1. Moved the original Figure S3 to the main text as the new Figure 2 with necessary narratives (Pages 9 and 10) to demonstrate that the existing DWBA method does not work satisfactorily in the coherent scattering cases where the surface patterns approach the coherent beam size. The new paragraph should orient the readers better before reading the FE-DWBA simulation subsection.

2. Added essential description of the FE-DWBA as suggested by the reviewer (Pages 11 and 12).
3. Addressing the reviewer's next few comments also helps to present both FE-DWBA and surface holography more coherently. We reformatted the text slightly by putting the FE-DWBA in its dedicated subsection, followed by the Holography subsection.

A lot of work has been done by the imaging community, CDI, Ptychography, Bragg ptychography, time dependent imaging etc. I did not find any reference to them. In fact, in my view a paragraph putting the current work in context to the imaging work will be important.

We focused mostly on reviewing existing work on dynamical scattering at surfaces and interfaces in the Introduction section. In this revision, we added a few sentences reviewing the current status of coherent X-ray imaging (Bragg CDI and ptychography) with an emphasis on the reconstruction methods based on kinematic approximations without taking the substrates into account (Page 3).

There are many instances where it is implied that CDI cannot work in case of multiple scattering (line 8). What is difficult to understand is that, if a real space image with nanometer resolution (at least tens of nm) is obtained, then that solves the structure problem. Or, are the authors claiming that the imaging community had not obtained the right image? Asked in a different way, what new structure information the current technique provide that cannot be obtained using CDI/Ptycho?

If we understand the reviewer's comment correctly, we may not have explained the unique nature of the multibeam scattering from substrate-supported planar but 3D patterns, where the substrate is part of the sample that causes the multi-beam scattering that supports the analyses of 3D information in a single view. In most current CDI and ptychography, samples are either free of or isolated from substrates and weakly interact with X-rays, as the 3D structure can be resolved but requires scanning numerous projection angles (Pages 3 and 4). Conventional CDI and Ptychography are not applicable to surface samples supported by substrates, which need to be measured in the reflection geometry and interpreted by appropriate dynamical scattering theory and simulation, such as the methods highlighted in this work (Pages 4 and 5).

An explanation of the origin of the intensity pattern will be very helpful. As it is, the paper quickly goes into analysis, but what general information is obtained from the complicated intensity pattern? After all, it is an experiment work, and as a result a clear explanation/understanding of data is important.

We intended to explain the scattering patterns around the reflected beam using the commonly known Born (kinematic) approximation in SI #1 and Figs. S1c and 1d. In the revision, mostly on Pages 8 and 9, we emphasized the geometrical transformation to correct the distorted q_x , q_y , and q_z directions due to the grazing incidence and reflection geometry. Also, we emphasized that the multi-beam scattering patterns due to the presence of the substrate are significantly different from those in the kinematic regime. We modified SI #1 and Fig. S1 to illustrate how the scattering pattern would have been treated if we had intended to solve the structure with

the kinematic approach. We also indicated more clearly that we focus our study in $\alpha_f - 2\theta$ space and $\alpha_f < 0.4^\circ$ where dynamical scattering dominates (Page 9). The added Fig. 2 and its narrative on DWBA (Pages 9 and 10) also serve the purpose of explaining the features in the coherent scattering pattern, which are important in the data interpretation and analysis.

The FE-DWBA is an interesting concept. Slicing of a thin film into several layers to fit reflectivity curves is well known. In this work the authors have applied similar idea but in all dimensions. By replacing height-height correlation by creating a grid, each one with a different refraction index, while maintaining the boundary conditions. A detailed explanation in the main text will be helpful. How is the initialization done? What are the constraints and how are they applied? Can the method work for a heterogeneous thin film? What is the computation cost of the method?

Following the suggestions by the reviewer, without significantly increasing the main article length, we added a paragraph that gives a more detailed explanation of how a 3D grid is generated (Page 11). We also included the technical details about the simulation and typical computation cost on Page 12. As we responded to Reviewer #2, the same method can be applied to heterogeneous thin films. However, the computation cost will be higher as fewer vertical stacks sharing the same electron density profile can be grouped (Page 14).

In the holography section, the first comment that I have is, as before, I found missing context. No referencing of holography or X-ray holography has been done. As mentioned above, both in surface scattering section and holography section, the referencing needs to improve. It should include current work done in hard and soft X-rays to inform readers of the state of the art in the field.

The review on X-ray holography was limited to a few successful and unsuccessful demonstrations of Lloyd's mirror concept in bright field mode with macroscopic mirrors, which have not been practical in solving nano- or micro-structures that are planar and supported by substrates. In the revision, we extended the review to holography in general and its state-of-the-art applications in the Introduction section (Page 6).

A hologram is a method whereby signal is mixed with a reference wave (usually plane wave) so that phase information is coded in. Thus, the beauty and strength of a hologram is in obtaining real space image. As best as I can understand, that is not the case here. In the present case, it looks like a two beam interference effect. Further, B is a point on the substrate? Shouldn't the scatter beam be significantly modified to be a reference beam to create a hologram. Why doesn't it create its own speckle pattern? From the data it looks like two co-propagating beam suffering similar scattering/reflection mechanism.

We agree with the reviewer that conventional holography using a plane wave as the reference beam is the simplest solution. Due to the high coherence, sensitivities to chemical and magnetic properties, and easiness of wavefront manipulation, most of the holographic methods have been developed in soft X-ray and EUV regimes. They often relied on computational

methods to retrieve the phase information for 2D or 3D real-space reconstruction. With hard X-rays, it is difficult to generate small and high-flux (and coherent) plane waves as the reference for studying nano- or micro-size samples. Thus far, the majority of hard X-ray holography uses the interference of X-ray-induced fluorescence for studying single-crystal structures. As suggested by the reviewer, a very brief review of holography is added to the manuscript in the introduction section (Page 6). To illustrate the holographic nature of our work, on Pages 16 and 17, we added a paragraph to highlight the interference terms more explicitly as follows. The interfering scattered waves can be simplified as

$$I(\alpha_i, \alpha_f, 2\theta) \sim \left| \int \rho(\mathbf{r}) \left\{ 1 + R(\alpha_f) e^{-2ik_z(\alpha_f)z} \right\} e^{-i\mathbf{q}\cdot\mathbf{r}} d\mathbf{r} \right|^2,$$

as the 1st term is the unperturbed scattering from the surface object and the 2nd term is the mirror image of the 1st term but modified by a complex reflection coefficient $R(\alpha_f)$ and the phase shift $2k_z(\alpha_f)z$ due to the mirroring effect by the total external reflection at the substrate surface (Path B, mirrored scattering), in the condition of a high incident angle (α_i) and low exit angle (α_f). With known substrate optical properties, the calculation of $R(\alpha_f)$ is relatively straightforward based on Parratt's algorithm, which also predicts a definitive constraint of $|R(\alpha_f = 0)| = 1$ with a phase shift of π in Path B. This constraint is shown in the holography images as $I(\alpha_i, \alpha_f = 0, 2\theta) = 0$, so that the intensity at the detector corresponding to the direction of the sample/substrate horizon is always 0, regardless of the incident angle and the sample structure. With the explanation, we hope we can convince the reviewer that our scattering geometry with the X-ray mirror substrate created a holographic setup.

Is it possible that at such low angle the whole sample is becoming an effective point scatterer due to "shortening" of the sample. Thus these fringes at long alpha but high 2theta are basically pseudo-Airy fringe type scattering?

Yes, the reviewer's comment is correct. At the low incident angles, the sample is projected to a shortened dimension along the X-ray beam direction. Around the reflected beam, pseudo-Airy patterns are evident and highlighted in SI #1 and Fig. S1. The scattering is from a pattern with 3D finite size of the rectangular prism or stacked rectangular prisms that give tale-tell information about the sample dimensions.

Will the method not be useful in case there is no multiple scattering? That makes the use-case very limited. A comment would be helpful.

The method is not needed in cases with no multiple scattering, e.g., specimens without a supporting substrate. Our methods are needed to solve the structure of substrate-supported samples in the cases such as critical-dimension metrology of integrated circuits where non-destructive measurement is critical. Furthermore, as demonstrated in this work, the presence of the substrates and the multiple scattering provides an opportunity to reveal the 3D information in a single-view image, which can be used to track changes of critical dimensions in the samples in real-time without resorting to the time-consuming angular scans. In addition,

the methods are applicable to the analyses of scattering by longer wavelength X-rays (e.g., EUV) and electron beams, where multiple scattering is prevalent.

Reviewer #2:

This manuscript reports on the development of a technique that uses the scattering of a doubly focused coherent X-ray beam under grazing incidence to characterize micro-nano-structures with large dimensions, of the order of those of the coherent beam. They show that, in this geometry, 2D images of the coherently scattered beam are very sensitive to every details of all the morphological features (shape and dimensions). They develop an approach to simulate the data, which is based on a combination of the well-known Distorted Wave Born Approximation (DWBA) and Parrat matrix used formalism to analyze X-ray reflectivity. This approach, although based on a first order approximation, is shown to account for every details on the morphology. This new approach is based on a Finite Element description of the sample, and a propagation, reflection and refraction of the X-ray waves between neighboring elements. With this so-called FE-DWBA description of coherent surface scattering, they demonstrate that each feature of the sample morphology yield characteristic features in the image. They remark that the images feature related to the sample height have a width twice smaller than expected for standard single scattering, and demonstrate that this is related to a holographic effect under grazing incidence in which coherent scattering between the object and its image with respect to the substrate plane interfere coherently. This is the so-called Lloyd's effect, which is well-known in the wavelengths range of radio waves, lasers and soft X-rays. They deduce that all the morphological information can be inferred from a single measurement a single grazing incidence angle larger than the critical angle for total external reflection, as opposed to more conventional coherent GISAXS (or coherent surface scattering), which requires measurements taken at different incident angles to properly analyze out-of plane features. The authors further demonstrate the extreme sensitivity of the technique to tiny variations of the morphology, on an object having a very small disorientation between two of its elements. By further neglecting the two minor scattering terms of the DWBA, the authors propose a much simplified version of the FE-DWBA analysis, which, like for GISAXS, corresponds approximately, if the incident angle is larger than the system's critical angles, to the mixing of the direct kinematically scattered wave and the wave scattered after its reflection by the substrate. Such a simulation does not require finite element-based numerical computation, and is thus much faster than the full FE-DWBA simulation.

They conclude that, with its 3D characterization capabilities, this new surface scattering method opens a wide field of characterization studies of objects supported by surfaces in a more effective way compared to other conventional coherent imaging techniques. Example of potential future studies include examples the metrology of 3D micro and nano –objects, the controlled fabrication of complex and multicomponent nanomaterials; the morphological characterization of thin-films photovoltaics or bio-membranes and supra-molecules in their aqueous environments etc.. Finally, they predict that, especially with the advance of novel 4th generation X-ray sources, such dynamic or holographic scattering will prove extremely useful for time-resolved in situ, in operando and possibly real-time measurements in realistic

sample environments.

This work establishes a new step forward in the capabilities of X-ray scattering, using a small focused, coherent beam under grazing incidence, to characterize a large diversity of systems, nondestructively, in very close to real conditions. They not only demonstrate the high potential of the technique, but also develop two new ways to analyze the data without resorting to full, extremely lengthy, full-fledged computational electromagnetics (CEM) techniques. This work is thus novel, and is based on further elaboration of previous works in the fields during the last decades.

We are extremely thankful to have the reviewer's very positive comments.

The work is detailed, very well presented, supported by detailed calculations and computational work, and supports most of its conclusions and claims. Its capability to fully characterize the complex morphology of a single object made of a single material with high density on a low-density substrate is clearly established. However, I would appreciate some discussion about its possible limitations in characterizing less straightforward objects such as multilayer, multicomponent, nano-patterned micro-nano electronics containing very diverse elements of different densities. I would also appreciate some discussion on the accuracy to which the sample dimensions could be deduced and how fluctuations or roughness of the edges of or interfaces within the sample could be taken into account.

The reviewer is correct that we used simple samples that have tale-tell scattering features to validate our computation methods and algorithms in both FE-DWBA and the first-principle calculation for the holography method. The advantage of FE-DWBA is that we use the full Parratt's reflectometry formulism in each grid cell, which is well-suited for multilayer and multicomponent surface samples. Without going into greater detail, we added discussions on more complex samples as suggested by the reviewer, for example, multilayer-based, multi-component, and diverse micro- and nano-electronics with reference to publications using soft-X-ray and EUV probes (Pages 14, 23, and 24). The accuracy of sample dimension evaluation is addressed in the new Fig. 6, where we demonstrate that the speckle size and phase depend on the sample dimensions with nm, even sub-nm sensitivity (Pages 21 and 22, Fig. 6). We appreciate the reviewer's comments about the edge and interfacial roughness of the mesoscale structures. In this work, we found that their most observable contribution is in the kinematic scattering regime, as shown in Fig. S1. The dominating effect of interfacial roughness is from the Au pattern surface, where the contrast is the greatest. The scattering is observed in the experiment data (Fig. 1a) and modeled in all simulations throughout the article (see Figs. 2c and 3c, Fig. S2b). The simulation of the surface-roughness scattering is based on Fourier transform of statistical roughness models. This is not the focus of this work and will be the subject of our future publications. We added a brief discussion in SI #1 and will not discuss it in the main article. The signature of edge roughness, on the other hand, is also observable in the higher exit angle ($\alpha_f > 0.4^\circ$) area in Fig. S1c (and other experimental scattering patterns) as the lost speckle contrast and fast diminishing scattering intensity in both q_x and q_y directions near the specular reflection. We did not try to simulate the contrast and intensity reduction effects, but we speculate that a similar method to model x-ray reflectivity data with the presence of

interfacial roughness can be used for quantifying the statistical distribution (correlation) of the edge roughness. This preliminary discussion may also address one of the questions raised by Reviewer #3 about 'the discrepancy between the simulations and experiment measurement near $2\theta = 0$ in Fig. 2c and Fig 3c'. Again, these phenomena are not closely related to the focus of this manuscript and will be the subject of our future publications on the high-resolution reconstruction of the scattering patterns based on Fourier transform methods.

The data interpretation, analysis and analysis development and discussion are built on firm grounds and are perfectly accurate. The conclusions are also strong and the work is shown to open a very wide field of studies. I would appreciate some more discussion and elaboration not only on the potential but also on the limits of the techniques, as just discussed above. The methodology is perfectly sounds and the work meets very high standards. The methods are provided with enough details for the work to be reproduced.

Again, we thank the reviewer's favorable comments. With the revision, we wish we have satisfactorily addressed the reviewer's concerns on the limitations of the methods, which are now discussed on **Pages 24 and 25**. The simulation methods and algorithm in this manuscript are limited to the forward calculation for illustrating multi-beam coherent surface scattering and holography and have not yet been extended to the reconstruction of the real-space structure from the scattering images. The feasibility of a full-fledged reconstruction incorporating the forward simulation is now briefly discussed on **Pages 23 to 25**. Even without a full reconstruction, we speculate that the most immediate application is to track changes of critical dimensions in the samples in real-time and non-destructive metrology of microelectronic circuits, where a priori information can be available from the design and complementary measurements. We also added an outlook of this work, including the possibility of developing algorithms to solve complex sample patterns with or without full reconstruction. This also partly addresses Reviewer #3's comments and questions.

Reviewer #3:

This paper presents two related ideas for simulating the scattering patterns in coherent surface scattering experiments. First, the authors introduced what they call FE-DWBA (finite-element distorted wave Born approximation) to address the coherence of the beam in coherent surface scattering measurements. Conventional DWBA does not work well when the sample length scales are about the size of the coherent beam. Secondly, based on their FE-DWBA results, they introduced a simpler version referred as the 'first-principle holography simulation' that reduces the complexity of the simulations to a situation more akin to Lloyds mirror interferometer of only 2 beams mixing.

The authors are highly commended for this work. Their work adds significant insight to the measured complex scattering patterns and is a notable contribution to this field. This work invites further research to refine and extend the general technique of coherent surface

scattering.

We appreciate very much the reviewer's complimentary remarks on our work.

However, a primary criticism this reviewer has is that in many places, the paper conflates techniques that can truly reconstruct a 3D image (eg, CSSI in ref. 22, or CDI or ptychography) with what they have done, which is the ability to calculate scattering patterns of a-priori known samples that resembles the measurements. An accurate statement about their discovery is the first sentence in the Conclusion and Discussion: 'In a most direct fashion, we demonstrated that coherent surface scattering in grazing incidence contains strong multiple scattering components, which encode the true 3D morphology of surface structures'. In other words, there is 3D information embedded in the measured scattering patterns. This is perfectly fine. But it does not imply, nor have they demonstrated, the ability to reconstruct the 3D sample, just from the measured scattering patterns. There are several statements in the text (see below) that suggests the ability to determine the 3D structure from a single coherent surface scattering pattern.

The reviewer noticed that we emphasized the demonstration of revealing the 3D morphology of surface structures through the FE-DWBA simulation and the first-principle holography calculations. As in our response to other reviewers also in this revision, we state clearly that the quantitative analysis is focused on the forward simulation based on *a priori* knowledge. Throughout the revised manuscript, we now clarify that we have not reached a stage of performing reconstruction using the single-view multi-beam coherent imaging. However, with both FE-DWBA and the first-principle method, our outlook is to explore the feasibility of reconstructing 3D using one or a limited number of single-view images, as also suggested by the reviewer.

The basic physics of this technique is that the substrate surface reflects a subset of the sample-scattered beam resulting in interference patterns which the authors have done an excellent job of simulating. However, the ability of this technique to provide true 3D morphology in a single-view has not been demonstrated and would seem to be extremely difficult without a-priori knowledge. Due to the geometry and substrate critical angle, each measurement only captures a subset of the sample scattering. For example, since the incident beam is $>$ critical angles, the near $q=0$ scattering will remain $>$ critical angle and so, will not be encoded in the interference patterns. That said, the technique could be useful in determining small 'in-operando' changes in morphology if the 'original' 3D structure of the sample was known a-priori.

The reviewer's comments on the determining small 'in operando' changes in morphology based on a priori information are appreciated. We agree with the reviewer that solving the 3D structure by rigorous reconstruction using only a single angle scattering image is difficult without a priori knowledge. However, the information in the specific scenario suggested by the reviewer is not completely lost in the single view scattering image. Without considering the multibeam scattering first, when the incident angle (α_i) is larger than the sample nominal

critical angles (θ_c), as the reviewer suggested, the near $\mathbf{q} = 0$ scattering impinges onto the substrate at an angle larger than θ_c and reflected by the substrate. The reflected scattering pattern is the Airy pattern of the sample around the so-called specular reflection (see Fig. S1c), similar to the scattering around $\mathbf{q} = 0$ as in transmission geometry, but $\mathbf{q} = [0, 0, q_z]$ with a finite q_z component of $2k\sin(\alpha_i)$ resulting from the substrate reflection, where $k = 2\pi/\lambda$ and $\alpha_i > \theta_c$. The scattering in this area can be treated in a kinematic approximation with an FT-based algorithm (Fig. S1d) as in Ref. [35], with which the 3D planar pattern can be reconstructed but requires scanning numerous incident angles. Most importantly, the kinematic approximation cannot explain the intense speckle patterns in the region where the exit angle (α_f) is between 0 (sample horizon) and θ_c , which should not have existed without the multi-beam scattering, the focus of this work. In this low region, the scattering propagates almost parallelly to the substrate no matter what the incident angle is. As we observed, the information carried by the speckle patterns is abundant about the sample form factors in all 3 dimensions. It would be our future work to evaluate whether the speckle patterns in the ‘forbidden’ area have enough oversampling rate for a full-fledged reconstruction. In the revision, we made these points more clearly in the discussion section (Pages 24 and 25), including referencing a recent work on reconstruction algorithm [Ref. 44] with conventional DWBA in the case of a weak in-plane perturbation to the in-plane electric field (see Pages 5 and 25). We thank the reviewer for provoking an interesting discussion here.

Some questions:

In the top 2 figures of Fig 2b are the wave-component assignments (TfTf and TfRf) flipped? If so, the same occurs in Fig S3.

Although the wave component assignments are not flipped in the original figures, we acknowledge that the illustration in the figures was somewhat incomplete and could be confusing. The figures are now improved by incorporating the final scattering directions. We thank the reviewer for pointing out the potentially confusing aspect of the figures.

Figure S3 caption has a reference to ‘Eq. (x)’

We fixed the error. The original Fig. S3 is now Fig. 2 in the main article.

Figure S2 caption first sentence suggests that simulations of BA and DWBA are shown, but in fact only the BA simulations are shown in Fig. S2. The DWBA simulations are shown in Fig. S3.

We thank the reviewer for spotting the error resulting from the reorganization of the main text and SI.

What causes the discrepancy between the simulations and experiment measurement near $2\theta=0$ in Fig 2c and Fig 3c? The simulations show strong fringes near $2\theta=0$ along the α direction but not seen in the measurement. Fig S4 explains the broad fringes for $\alpha < 0.4$ deg, but does not say anything about the finer fringes between 0.6 deg $> \alpha > 0.4$

deg.

The fringes below the reflected beam spot that the reviewer referred to are the Airy patterns of the sample propagating in the x-direction (X-ray direction) as shown in **Figs. S1c** (experimental data) and **S1d** (simulation) in a magnified view. The mismatch in intensity (simulation vs. experimental data) is from 3 factors: 1) an artifact from the data collection when we used an attenuator in front of the detector to reduce the intense reflected beam so we could make the exposure time long enough to obtain a sufficient signal-to-noise ratio for the fringes for $\alpha_f < 0.4$ deg, the focus of the experiment. 2) The edge roughness of the sample can reduce the fringe contrast and intensity, also, as in response to Reviewer #2. 3) The detector with the 75- μm pixel size at the 5-m position just barely resolves the 4- μm bar width. **Comments are added to SI #1** to explain the discrepancy, but not in other figures and the main text, for brevity.

In the discussion on applications, the authors mention biological membranes and supra-molecules in aqueous environments. Seems like a very long stretch. Membranes scatter very weakly and tend to have unknown complex folds.

We recognize that biomembranes and supramolecules, especially in aqueous environments, do not have high scattering contrast. These could be the most challenging but very interesting substrate-supported systems to study. The grazing-incident surface scattering benefits from the surface enhancement that may be better capable than conventional X-ray techniques of imaging the mesoscale features in the multi-fold structure (**Page 26**). We think we should keep this speculation to inspire future research in the area. Also, we should be able to take advantage of a priori information during *in situ* measurements.

Given this improved ‘forward modeling’, have the authors tried to implement them (with multiple views) into the iterative reconstruction algorithms used in ref [22] ? Perhaps it would lead to improved reconstructions?

This is a great suggestion. In Fig. S3 we showed the scattering pattern at three incident angles. We are in the process of working on the patterns and seeking improvement in simulation, which may lead to a reconstruction similar to the work in the original Ref. 22 (now 35), but with a much higher resolution. We added a comment in **SI #4**.

Examples of not quite accurate statements:

Abstract, first sentence: ‘Visualizing surface supported and buried planar mesoscale structures such as nanoelectronics, ultrathin-film quantum dots, photovoltaics, and heterogeneous catalysts, require high-resolution imaging with coherent surface X-ray scattering’. This statement is clearly not accurate – imaging of nanoelectronics has been performed using ptychography and transmissions X-ray microscopy, both of which are not ‘coherent surface X-ray scattering’.

We took the reviewer's suggestions and revised the sentence in the abstract accordingly (Page 1).

Abstract, second sentence: 'Here, we discover that the dynamical or multibeam scattering n grazing-incident reflection geometry promises three-dimensional structural determination in a single view but cannot be reconstructed by the conventional Fourier-transform approaches'. As mentioned above, this sentence is quite a stretch.

Although we did not claim that we could determine structures in a single view by reconstruction, we revised the sentence, so it implies less about 'determination'. (Page 1)

Abstract, last sentence: 'The holographic imaging paves the way for single-shot 3D structure determination...'. As discussed above, this sentence is not accurate.

We respectfully argue that the phrase 'paves the way' is acceptable here for the following reasons. First, based on Eq. 2 in the revision, the two-term scattering pattern cannot be calculated simply using the conventional FT algorithm. However, both FE-DWBA and the holographic simulation provide viable methods for forward calculation from real space to reciprocal space, one of the most important steps for reconstruction. Second, the scattering images also contain a vast amount of kinematic scattering signal, which can be used to either model or reconstruct real-space patterns that, in turn, assist the forward simulation. We did not claim it can be done now, but what we demonstrated here would be a necessary step to understand the feasibility of model-independent reconstruction.

Page 10: 'The relationship between the oscillationbecause the sample's 3D structure can now be determined by a single scattering pattern at a single incident angle.' As discussed above, this sentence is not accurate.

We acknowledge that the 3D structure determination is not from model-independent reconstruction; rather, we used model-dependent forward calculation. Since every dimension of the sample has its signature in the holography pattern, the quantitative forward calculation allowed us to determine the dimensions accurately (also see next point). We slightly modified the sentence and other parts of the text to reflect the fact that the dimension determination is achieved by using the forward simulation (e.g., on Pages 13 and 25).

In two places (eg, second to last sentence in Abstract, and second to last sentence in Introduction), the authors mention 'nanometer resolutions' and 'sub-nm spatial resolution' respectively. They suggest that the scattering patterns can reveal such resolutions. There is no data to support these claims. The simulations in Fig 4 (h,i,m,n) do not support such claims.

We thank the reviewer for pointing out this important issue. In the revision, we added a new figure (Fig. 6) and necessary narratives on Pages 21 and 22 to address the spatial sensitivity issues. Using simulation results with parameter variations much finer than those used in the original Figs. 4h,i, m,and n, we illustrated that the sensitivity of the scattering patterns could

reveal detectable changes in the speckle patterns due to a 3-nm change in the top bar width and a 1-nm change in its thickness. The values are better than what the detector range promises because of the oversampled speckles. The results also address Reviewer #2's question on the accuracy of the sample dimension evaluation.

In summary, I would recommend this paper for publication in Nature Communications after a careful and substantive text edit. The work is technically sound and appropriate for Nature Communications, but the text needs to be improved. This is a highly commendable technical piece of work and it deserves to be published and read – in a clear and un-inflated narrative.

Again, we appreciate the reviewer's positive remarks. With the revision, we hope we satisfactorily addressed the reviewer's concerns and criticism.

REVIEWERS' COMMENTS

Reviewer #1 (Remarks to the Author):

The authors have revised the manuscript. I think the work is of high standard and rigorous. Combining two advanced techniques is non trivial and the authors have achieved that. I think the paper is in publishable form.

Reviewer #2 (Remarks to the Author):

I am fully satisfied by the author's answers, and consider that the manuscript is suitable for publication in Nature Communications in its present form.

Reviewer #3 (Remarks to the Author):

The authors have made all the necessary edits and I strongly recommend publication of their paper.

Reviewer #1 (Remarks to the Author):

The authors have revised the manuscript. I think the work is of high standard and rigorous. Combining two advanced techniques is non trivial and the authors have achieved that. I think the paper is in publishable form.

Our response: Once again, we appreciate very much the reviewer's recognition of our work and the valuable comments that helped us to improve the presentation of the manuscript.

Reviewer #2 (Remarks to the Author):

I am fully satisfied by the author's answers, and consider that the manuscript is suitable for publication in Nature Communications in its present form.

Our response: We thank the reviewer's favorable comments. We acknowledge that the reviewer's previous comments and suggestions guided us to give serious and careful thought to the immediate applications of the work.

Reviewer #3 (Remarks to the Author):

The authors have made all the necessary edits and I strongly recommend publication of their paper.

Our response: We thank the reviewer's favorable recommendations. Also, the reviewer's previous comments were valuable for us in improving the narrative of the results.